# Counting Distinct Elements Under Person-Level Differential Privacy

**Alexander Knop**[*]
Google
alexanderknop@google.com

**Thomas Steinke**[*]
Google DeepMind
steinke@google.com

## Abstract

We study the problem of counting the number of distinct elements in a dataset subject to the constraint of differential privacy. We consider the challenging setting of person-level DP (a.k.a. user-level DP) where each person may contribute an unbounded number of items and hence the sensitivity is unbounded.

Our approach is to compute a bounded-sensitivity version of this query, which reduces to solving a max-flow problem. The sensitivity bound is optimized to balance the noise we must add to privatize the answer against the error of the approximation of the bounded-sensitivity query to the true number of unique elements.

## 1 Introduction

An elementary data analysis task is to count the number of distinct elements occurring in a dataset. The dataset may contain private data and even simple statistics can be combined to leak sensitive information about people [DN03]. Our goal is to release (an approximation to) this count in a way that ensures the privacy of the people who contributed their data. As a motivating example, consider a collection of internet browsing histories, in which case the goal is to compute the total number of websites that have been visited by at least one person.

Differential privacy (DP) [DMNS06] is a formal privacy standard. The simplest method for ensuring DP is to add noise (from either a Laplace or Gaussian distribution) to the true answer, where the scale of the noise corresponds to the sensitivity of the true answer – i.e., how much one person's data can change the true value.

If each person contributes a single element to the dataset, then the sensitivity of the number of unique elements is one. However, a person may contribute multiple elements to the dataset and our goal is to ensure privacy for all of these contributions simultaneously. That is, we seek to provide person-level DP (a.k.a. user-level DP[2]).

This is the problem we study: We have a dataset $D = (u_1, u_2, \cdots, u_n)$ of person records. Each person $i \in [n]$ contributes a finite dataset $u_i \in \Omega^*$, where $\Omega$ is some (possibly infinite) universe of potential elements (e.g., all finite-length binary strings) and $\Omega^* := \bigcup_{\ell \in \mathbb{N}} \Omega^\ell$ denotes all subsets of $\Omega$ of finite size. Informally, our goal is to compute the number of unique elements

$$\mathrm{DC}(D) := \left| \bigcup_{i \in [n]} u_i \right| \tag{1}$$

---

[*]Alphabetical author order.

[2]We prefer the term "person" over "user," as the latter only makes sense in some contexts and could be confusing in others.

37th Conference on Neural Information Processing Systems (NeurIPS 2023).

in a way that preserves differential privacy. A priori, the sensitivity of this quantity is infinite, as a single person can contribute an unbounded number of unique elements.

In particular, it is not possible to output a meaningful upper bound on the number of distinct elements subject to differential privacy. This is because a single person could increase the number of distinct elements arbitrarily and differential privacy requires us to hide this contribution. It follows that we cannot output a differentially private unbiased estimate of the number of distinct elements with finite variance. However, it is possible to output a lower bound. Thus our formal goal is to compute a high-confidence lower bound on the number of distinct elements that is as large as possible and which is computed in a differentially private manner.

## 1.1  Our Contributions

Given a dataset $D = (u_1, \cdots, u_n) \in (\Omega^*)^n$ and an integer $\ell \geq 1$, we define

$$\mathrm{DC}(D; \ell) := \max \left\{ \left| \bigcup_{i \in [n]} v_i \right| : \forall i \in [n] \ \ v_i \subseteq u_i \wedge |v_i| \leq \ell \right\}. \tag{2}$$

That is, $\mathrm{DC}(D; \ell)$ is the number of distinct elements if we restrict each person's contribution to $\ell$ elements. We take the maximum over all possible restrictions.

It is immediate that $\mathrm{DC}(D; \ell) \leq \mathrm{DC}(D)$ for all $\ell \geq 1$. Thus we obtain a lower bound on the true number of unique elements. The advantage of $\mathrm{DC}(D; \ell)$ is that its sensitivity is bounded by $\ell$ (see Lemma A.1 for a precise statement) and, hence, we can estimate it in a differentially private manner. Specifically,

$$\mathcal{M}_{\ell, \varepsilon}(D) := \mathrm{DC}(D; \ell) + \mathrm{Lap}\left(\ell / \varepsilon\right) \tag{3}$$

defines an $\varepsilon$-DP algorithm $M_{\ell, \varepsilon} : (\Omega^*)^n \to \mathbb{R}$, where $\mathrm{Lap}\left(b\right)$ denotes Laplace noise scaled to have mean $0$ and variance $2b^2$. This forms the basis of our algorithm. Two challenges remain: Setting the sensitivity parameter $\ell$ and computing $\mathrm{DC}(D; \ell)$ efficiently.

To obtain a high-confidence lower bound on the true distinct count, we must compensate for the Laplace noise, which may inflate the reported value. We can obtain such a lower bound from $\mathcal{M}_\ell(D)$ using the cumulative distribution function (CDF) of the Laplace distribution: That is, $\forall b > 0 \ \forall \beta \in (0, 1/2] \ \mathbb{P}\left[\mathrm{Lap}\left(b\right) \geq b \cdot \log\left(\frac{1}{2\beta}\right)\right] = \beta$, so

$$\mathbb{P}\left[\underbrace{\mathcal{M}_{\ell, \varepsilon}(D) - \frac{\ell}{\varepsilon} \cdot \log\left(\frac{1}{2\beta}\right) \leq \mathrm{DC}(D)}_{\text{lower bound}}\right] \geq \underbrace{1 - \beta}_{\text{confidence}}. \tag{4}$$

**Choosing the sensitivity parameter $\ell$.**  Any choice of $\ell \geq 1$ gives us a lower bound: $\mathrm{DC}(D; \ell) \leq \mathrm{DC}(D)$. Since $\forall D \ \lim_{\ell \to \infty} \mathrm{DC}(D; \ell) = \mathrm{DC}(D)$, this lower bound can be arbitrarily tight. However, the larger $\ell$ is, the larger the sensitivity of $\mathrm{DC}(D; \ell)$ is. That is, the noise we add scales linearly with $\ell$.

Thus there is a bias-variance tradeoff in the choice of $\ell$. To make this precise, suppose we want a lower bound on $\mathrm{DC}(D)$ with confidence $1 - \beta \in [\frac{1}{2}, 1)$, as in Equation (4). To obtain the tightest possible lower bound with confidence $1 - \beta$, we want $\ell$ to maximize the expectation

$$q(D; \ell) := \mathrm{DC}(D; \ell) - \frac{\ell}{\varepsilon} \cdot \log\left(\frac{1}{2\beta}\right) = \mathbb{E}_{\mathcal{M}_{\ell, \varepsilon}}\left[\mathcal{M}_{\ell, \varepsilon}(D) - \frac{\ell}{\varepsilon} \cdot \log\left(\frac{1}{2\beta}\right)\right]. \tag{5}$$

We can use the exponential mechanism [MT07] to privately select $\ell$ that approximately maximizes $q(D; \ell)$. However, directly applying the exponential mechanism is problematic because each score has a different sensitivity – the sensitivity of $q(\cdot; \ell)$ is $\ell$. Instead, we apply the Generalized Exponential Mechanism (GEM) of Raskhodnikova and Smith [RS15] (see Algorithm 3). Note that we assume some a priori maximum value of $\ell$ is supplied to the algorithm; this is $\ell_{\max}$.

Our main algorithm attains the following guarantees.

**Theorem 1.1** (Theoretical Guarantees of Our Algorithm). *Let $\varepsilon > 0$ and $\beta \in (0, \frac{1}{2})$ and $\ell_{\max} \in \mathbb{N}$. Define $\mathcal{M} : (\Omega^*)^* \to \mathbb{N} \times \mathbb{R}$ to be $\mathcal{M}(D) = \mathrm{DPDISTINCTCOUNT}(D; \ell_{\max}, \varepsilon, \beta)$ from Algorithm 1. Then $\mathcal{M}$ satisfies all of the following properties.*

- **Privacy:** *$\mathcal{M}$ is $\varepsilon$-differentially private.*

- **Lower bound:** *For all $D \in (\Omega^*)^n$,*

$$\mathbb{P}_{(\hat{\ell}, \hat{\nu}) \leftarrow \mathcal{M}(D)} [\hat{\nu} \leq \mathrm{DC}(D)] \geq 1 - \beta. \tag{6}$$

- **Upper bound:** *For all $D \in (\Omega^*)^n$,*

$$\mathbb{P}_{(\hat{\ell}, \hat{\nu}) \leftarrow \mathcal{M}(D)} \left[ \hat{\nu} \geq \max_{\ell \in [\ell_{\max}]} \mathrm{DC}(D; \ell) - \frac{10\ell + 18\ell_A^*}{\varepsilon} \log\left(\frac{\ell_{\max}}{\beta}\right) \right] \geq 1 - 2\beta, \tag{7}$$

*where $\ell_A^* = \arg\max_{\ell \in [\ell_{\max}]} \mathrm{DC}(D; \ell) - \frac{\ell}{\varepsilon} \log\left(\frac{1}{2\beta}\right)$.*

- **Computational efficiency:** *$\mathcal{M}(D)$ has running time $O\left(|D|^{1.5} \cdot \ell_{\max}^2\right)$, where $|D| := \sum_i |u_i|$.*

The upper bound guarantee (7) is somewhat difficult to interpret. However, if the number of items per person is bounded by $\ell_*$, then we can offer a clean guarantee: If $D = (u_1, \cdots, u_n) \in (\Omega^*)^n$ satisfies $\max_{i \in [n]} |u_i| \leq \ell_* \leq \ell_{\max}$, then combining the upper and lower bounds of Theorem 1.1 gives

$$\mathbb{P}_{(\hat{\ell}, \hat{\nu}) \leftarrow \mathcal{M}(D)} \left[ \mathrm{DC}(D) \geq \hat{\nu} \geq \mathrm{DC}(D) - \frac{28\ell_*}{\varepsilon} \log\left(\frac{\ell_{\max}}{\beta}\right) \right] \geq 1 - 3\beta. \tag{8}$$

Note that $\ell_*$ is not assumed to be known to the algorithm, but the accuracy guarantee is able to adapt. We only assume $\ell_* \leq \ell_{\max}$, where $\ell_{\max}$ is the maximal sensitivity considered by the algorithm.

In addition to proving the above theoretical guarantees, we perform an experimental evaluation of our algorithm.

---

**Algorithm 1** Distinct Count Algorithm

---

1: **procedure** SENSITIVEDISTINCTCOUNT($D = (u_1, \cdots, u_n) \in (\Omega^*)^n; \ell \in \mathbb{N}$)      $\triangleright \mathrm{DC}(D; \ell)$
2:      Let $U_\ell = \bigcup_{i \in [n]} \left(\{i\} \times [\min\{\ell, |u_i|\}]\right) \subset [n] \times [\ell]$.
3:      Let $V = \bigcup_{i \in [n]} u_i \subset \Omega$.
4:      Define $E_\ell \subseteq U \times V$ by $((i,j), v) \in E \iff v \in u_i$.
5:      Let $G_\ell$ be a bipartite graph with vertices partitioned into $U_\ell$ and $V$ and edges $E_\ell$.
6:      $m_\ell \leftarrow$ MAXIMUMMATCHINGSIZE($G$).      $\triangleright$ [HK73; Kar73]
7:      **return** $m_\ell \in \mathbb{N}$
8: **end procedure**
9: **procedure** DPDISTINCTCOUNT($D = (u_1, \cdots, u_n) \in (\Omega^*)^n; \ell_{\max} \in \mathbb{N}, \varepsilon > 0, \beta \in (0, \frac{1}{2})$)
10:      **for** $\ell \in [\ell_{\max}]$ **do**
11:          Define $q_\ell(D) := $ SENSITIVEDISTINCTCOUNT($D; \ell$) $- \frac{2\ell}{\varepsilon} \cdot \log\left(\frac{1}{2\beta}\right)$.
12:      **end for**
13:      $\hat{\ell} \leftarrow$ GEM($D; \{q_\ell\}_{\ell \in [\ell_{\max}]}, \{\ell\}_{\ell \in [\ell_{\max}]}, \varepsilon/2, \beta$).      $\triangleright$ Algorithm 3
14:      $\hat{\nu} \leftarrow q_{\hat{\ell}}(D) + \mathrm{Lap}\left(2\hat{\ell}/\varepsilon\right)$.
15:      **return** $(\hat{\ell}, \hat{\nu}) \in [\ell_{\max}] \times \mathbb{R}$.
16: **end procedure**

---

**Efficient computation.** The main computational task for our algorithm is to compute $\mathrm{DC}(D; \ell)$. By definition (2), this is an optimization problem. For each person $i \in [n]$, we must select a subset $v_i$ of that person's data $u_i$ of size at most $\ell$ so as to maximize the size of the union of the subsets $\left| \bigcup_{i \in [n]} v_i \right|$.

We can view the dataset $D = (u_1, \cdots, u_n) \in (\Omega^*)^n$ as a bipartite graph. On one side we have the $n$ people and on the other side we have the elements of the data universe $\Omega$.[3] There is an edge between $i \in [n]$ and $x \in \Omega$ if and only if $x \in u_i$.

We can reduce computing $\mathrm{DC}(D; \ell)$ to a max-flow problem: Each edge in the bipartite graph has capacity one. We add a source vertex $s$ which is connected to each person $i \in [n]$ by an edge with capacity $\ell$. Finally we add a sink $t$ that is connected to each $x \in \Omega$ by an edge with capacity 1. The max flow through this graph is precisely $\mathrm{DC}(D; \ell)$.

Alternatively, we can reduce computing $\mathrm{DC}(D; \ell)$ to bipartite maximum matching. For $\ell = 1$, $\mathrm{DC}(D; 1)$ is exactly the maximum cardinality of a matching in the bipartite graph described above. For $\ell \geq 2$, we simply create $\ell$ copies of each person vertex $i \in [n]$ and then $\mathrm{DC}(D; \ell)$ is the maximum cardinality of a matching in this new bipartite graph.[4]

Using this reduction, standard algorithms for bipartite maximum matching [HK73; Kar73] allow us to compute $\mathrm{DC}(D; \ell)$ with $O(|D|^{1.5} \cdot \ell)$ operations. We must repeat this computation for each $\ell \in [\ell_{\max}]$.

---

**Algorithm 2** Linear-Time Approximate Distinct Count Algorithm

---

1: **procedure** DPAPPROXDISTINCTCOUNT($D = (u_1, \cdots, u_n) \in (\Omega^*)^n$; $\ell_{\max} \in \mathbb{N}$, $\varepsilon > 0$, $\beta \in (0, \frac{1}{2})$)
2: $\quad$ $S \leftarrow \emptyset$.
3: $\quad$ **for** $\ell \in [\ell_{\max}]$ **do**
4: $\quad\quad$ **for** $i \in [n]$ with $u_i \setminus S \neq \emptyset$ **do**
5: $\quad\quad\quad$ Choose lexicographically first $v \in u_i \setminus S$. $\qquad\qquad$ ▷ Match $(i, \ell)$ to $v$.
6: $\quad\quad\quad$ Update $S \leftarrow S \cup \{v\}$.
7: $\quad\quad$ **end for**
8: $\quad\quad$ Define $q_\ell(D) := |S| - \frac{2\ell}{\varepsilon} \cdot \log\left(\frac{1}{2\beta}\right)$. $\qquad$ ▷ This loop computes $\{q_\ell(D)\}_{\ell \in [\ell_{\max}]}$.
9: $\quad$ **end for**
10: $\quad$ $\hat{\ell} \leftarrow \mathrm{GEM}(D; \{q_\ell\}_{\ell \in [\ell_{\max}]}, \{\ell\}_{\ell \in [\ell_{\max}]}, \varepsilon/2, \beta)$. $\qquad$ ▷ Algorithm 3
11: $\quad$ $\hat{\nu} \leftarrow q_{\hat{\ell}}(D) + \mathrm{Lap}\left(2\hat{\ell}/\varepsilon\right)$.
12: $\quad$ **return** $(\hat{\ell}, \hat{\nu}) \in [\ell_{\max}] \times \mathbb{R}$.
13: **end procedure**

---

**Linear-time algorithm.** Our algorithm above is polynomial-time. However, for many applications the dataset size $|D|$ is enormous. Thus we also propose a linear-time variant of our algorithm. However, we must trade accuracy for efficiency.

There are two key ideas that differentiate our linear-time algorithm (Algorithm 2) from our first algorithm (Algorithm 1) above: First, we compute a maxim*al* bipartite matching instead of a maxim*um* bipartite matching.[5] This can be done using a linear-time greedy algorithm and gives a 2-approximation to the maximum matching. (Experimentally we find that the approximation is better than a factor of 2.) Second, rather than repeating the computation from scratch for each $\ell \in [\ell_{\max}]$, we incrementally update our a maximal matching while increasing $\ell$. The main challenge here is ensuring that the approximation to $\mathrm{DC}(D; \ell)$ has low sensitivity – i.e., we must ensure that our approximation algorithm doesn't inflate the sensitivity. Note that $\mathrm{DC}(D; \ell)$ having low sensitivity does not automatically ensure that the approximation to it has low sensitivity.

**Theorem 1.2** (Theoretical Guarantees of Our Linear-Time Algorithm). *Let* $\varepsilon > 0$ *and* $\beta \in (0, \frac{1}{2})$ *and* $\ell_{\max} \in \mathbb{N}$. *Define* $\mathcal{M} : (\Omega^*)^* \rightarrow \mathbb{N} \times \mathbb{R}$ *to be* $\widehat{\mathcal{M}}(D) =$

---

[3] The data universe $\Omega$ may be infinite, but we can restrict the computation to the finite set $\bigcup_{i \in [n]} u_i$. Thus there are at most $n + \mathrm{DC}(D) \leq n + |D|$ item vertices in the graph.

[4] We need only create $\min\{\ell, |u_i|\}$ copies of the person $i \in [n]$. Thus the number of person vertices is at most $\min\{n\ell, |D|\}$.

[5] To clarify the confusing terminology: A matching is a subset of edges such that no two edges have a vertex in common. A maximum matching is a matching of the largest possible size. A maximal matching is a matching such that no edge could be added to the matching without violating the matching property. A maximum matching is also a maximal matching, but the reverse is not true.

| Data Set | Vocabulary Size | Estimated Vocabulary Size | | |
|---|---|---|---|---|
| | | 10th Percentile | Median | 90th Percentile |
| Amazon Fashion | 1450 | 1220.6 | 1319.1 | 1394.2 |
| Amazon Industrial and Scientific | 36665 | 35970.5 | 36198.9 | 36326.7 |
| Reddit | 102835 | 102379.7 | 102512.6 | 102643.9 |
| IMDB | 98726 | 98555.6 | 98670.4 | 98726.8 |

Table 1: True and estimated (using DPDistinctCount with $\varepsilon = 1$, $\beta = 0.05$ and $\ell_{\max} = 100$) counts per data set.

DPAPPROXDISTINCTCOUNT$(D; \ell_{\max}, \varepsilon, \beta)$ *from Algorithm 2. Then $\widehat{\mathcal{M}}$ satisfies all of the following properties.*

- ***Privacy:*** *$\widehat{\mathcal{M}}$ is $\varepsilon$-differentially private.*

- ***Lower bound:*** *For all $D \in (\Omega^*)^n$,*

$$\mathbb{P}_{(\hat{\ell}, \hat{\nu}) \leftarrow \widehat{\mathcal{M}}(D)} [\hat{\nu} \leq \mathrm{DC}(D)] \geq 1 - \beta. \tag{9}$$

- ***Upper bound:*** *If $D = (u_1, \cdots, u_n) \in (\Omega^*)^n$ satisfies $\max_{i \in [n]} |u_i| \leq \ell_* \leq \ell_{\max}$, then*

$$\mathbb{P}_{(\hat{\ell}, \hat{\nu}) \leftarrow \widehat{\mathcal{M}}(D)} \left[ \hat{\nu} \geq \frac{1}{2} \mathrm{DC}(D) - \frac{28\ell_*}{\varepsilon} \log \left( \frac{\ell_{\max}}{\beta} \right) \right] \geq 1 - 2\beta. \tag{10}$$

- ***Computational efficiency:*** *$\mathcal{M}(D)$ has running time $O\left(|D| + \ell_{\max} \log \ell_{\max}\right)$, where $|D| := \sum_i |u_i|$.*

The factor $\frac{1}{2}$ in the upper bound guarantee (10) is the main loss compared to Theorem 1.1. (The win is $O(|D|)$ runtime.) This is a worst-case bound and our experimental result show that for realistic data the performance gap is not so bad.

The proofs of Theorems 1.1 and 1.2 are in Appendix A.

## 2 Related Work

Counting the number of distinct elements in a collection is one of the most fundamental database computations. This is supported as the COUNT(DISTINCT ...) operation in SQL. Hence, unsurprisingly, the problem of computing the number of unique elements in a differentially private way has been extensively investigated.

In the case where we assume each person contributes only one element (a.k.a. event-level privacy or item-level privacy), the number of distinct elements has sensitivity 1 and, hence, we can simply use Laplace (or Gaussian) noise addition to release. However, it may not be possible to compute the number of distinct elements exactly due to space, communication, or trust constraints (e.g. in the local model of DP [KLNRS11]).

Most efforts have been focused on creating differentially private algorithms for counting distinct elements under space constraints (and assuming each person contributes a single element). To save space, we wish to compute a small summary of the dataset (called a sketch) that allows us to estimate the number of distinct elements and which can be updated as more elements are added. Smith, Song, and Thakurta [SST20] proved that a variant of the Flajolet-Martin sketch is private and Pagh and Stausholm [PS20] analyzed a sketch over the binary finite field. Dickens, Thaler, and Ting [DTT22] proved a general privacy result for order-invariant cardinality estimators. Hehir, Ting, and Cormode [HTC23] provided a mergeable private sketch (i.e. two sketches can be combined to obtain a sketch of the union of the two datasets). In contrast, Desfontaines, Lochbihler, and Basin [DLB19] proved an impossibility result for mergeable sketches, which shows that privacy or accuracy must degrade as we merge sketches.

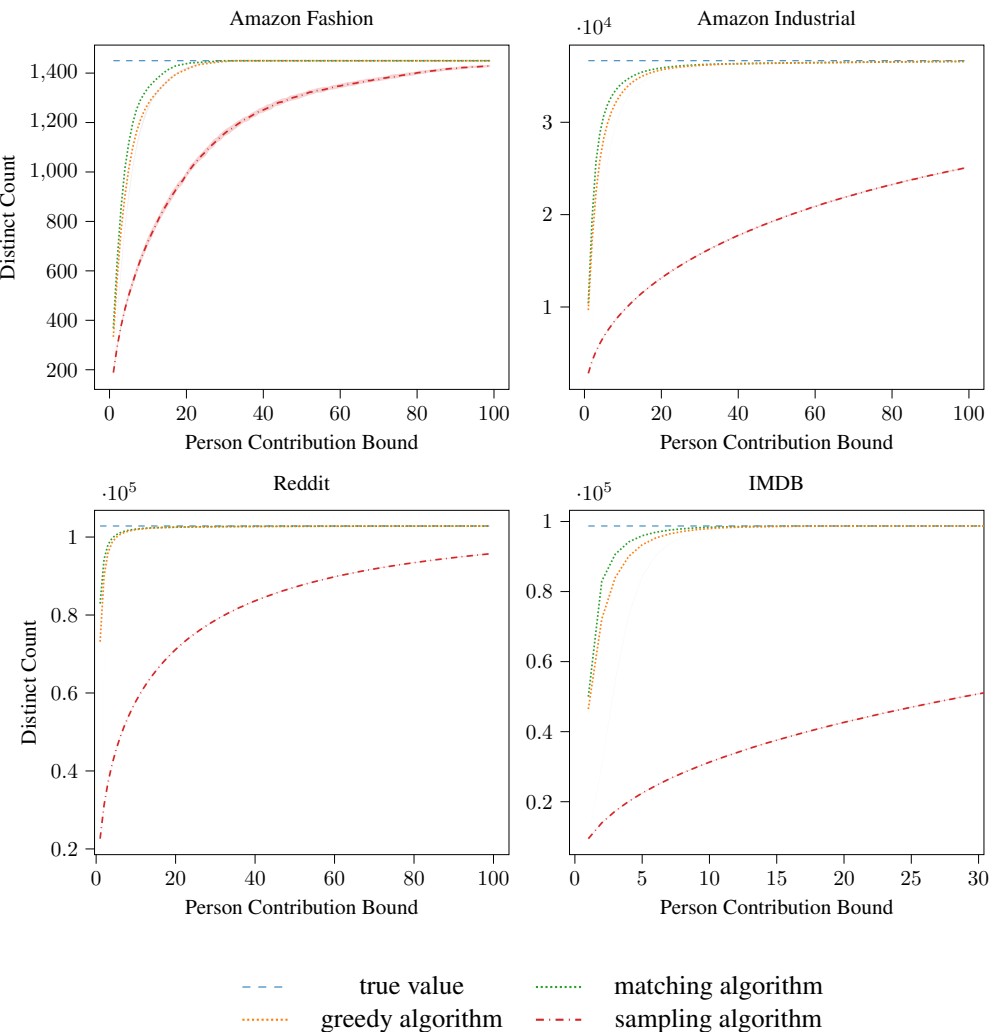

Figure 1: Performance of different algorithms estimating distinct count assuming that each person can contribute at most $\ell$ elements (e.g., these algorithms are estimating $\mathrm{DC}(D; \ell)$). (These algorithms have bounded sensitivity, but we do not add noise for privacy yet.)

Counting unique elements has been considered in the pan-private streaming setting [DNPRY10] (the aforementioned algorithms also work in the pan-private setting) and in the continual release streaming setting [GKNM23]. (In the continual release setting the approximate count is continually updated, while in the pan-private setting the approximate count is only revealed once, but at an unknown point in time.) Kreuter, Wright, Skvortsov, Mirisola, and Wang [KWSMW20] give private algorithms for counting distinct elements in the setting of secure multiparty computation. In the local and shuffle models, the only known results are communication complexity bounds [CGKM21].

A closely related problem is that of identifying as many elements as possible (rather than just counting them); this is known as "partition selection," "set union," or "key selection" [SDH23; DVGM22; KKMN09; CWG22; RCR22; GGKSSY20; ZDKSTMAS23]. Note that, by design, DP prevents us from identifying elements that only appear once in the dataset, or only a few times. Thus we can only output items that appear frequently.

The most closely related work to ours is that of Dong, Fang, Yi, Tao, and Machanavajjhala [DFYTM22] and Fang, Dong, and Yi [FDY22]. These papers present two different algorithms for privately approximating the distinct count (and other statistics). We discuss these below and present an experimental comparison in Table 2. We also remark that both papers prove instance optimality guarantees for their algorithms.

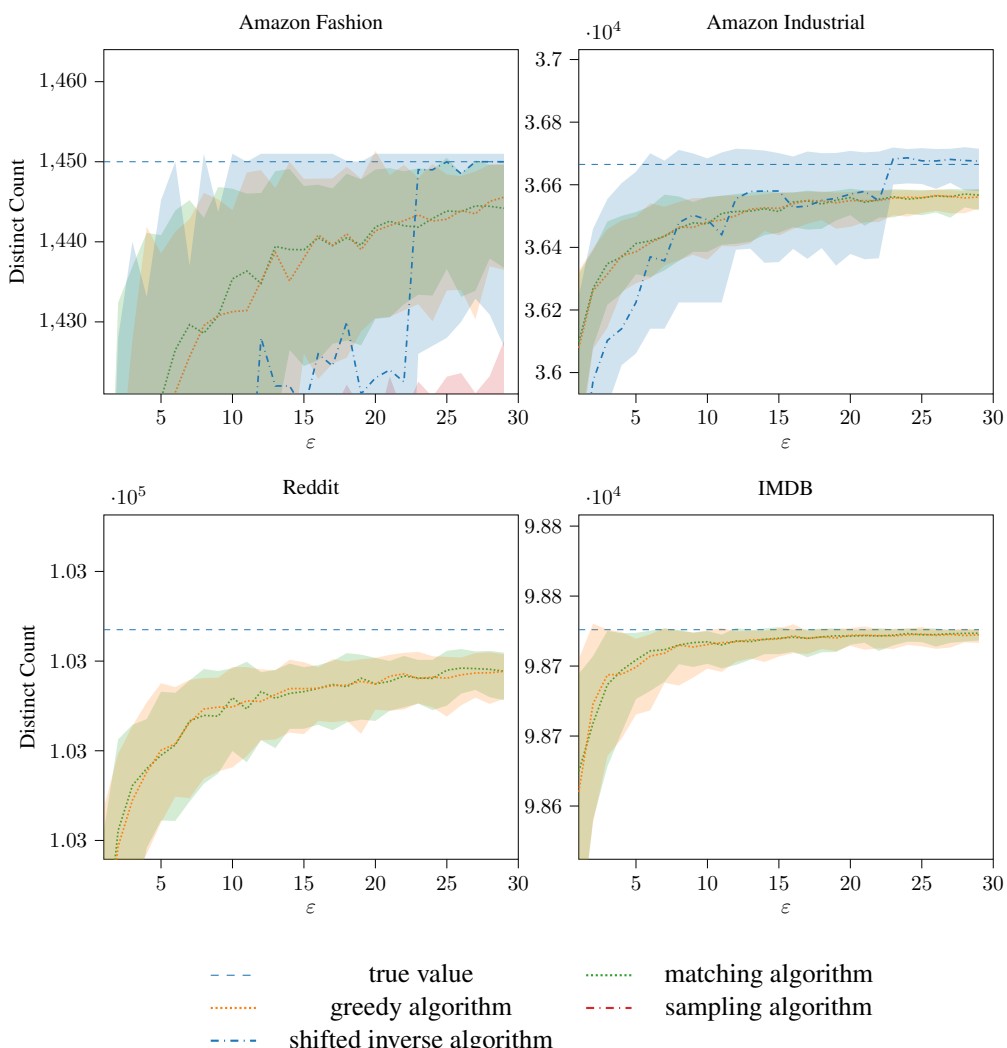

Figure 2: Performance of different algorithms estimating distinct count in a differentially private way for different values of $\varepsilon$; for all of them $\beta = 0.05$ and $\ell_{\max} = 100$. The values between 10th and 90th percentile of each algorithms estimation are shaded into corresponding colors. For the shifted inverse algorithm, the first two plots contain the results for $\beta = 0.05$ and $D$ equal to the true number of distinct elements in the dataset. The later two datasets are lacking the results for shifted inverse algorithm due to the computational constraints.

| User | Supplier | | Customer | |
| --- | --- | --- | --- | --- |
| Attribute | PS.AQ | L.EP | O.OD | L.RD |
| R2T [DFYTM22] | 0.0658 | 0.1759 | 0.0061 | 0.150 |
| (Approx)ShiftedInverse [FDY22] | 0.0553 | 0.0584 | 0.005 | 0.0061 |
| DPApproxDistinctCount | 0.0140 | 0.0110 | 0.0008 | 0.0037 |
| DPDistinctCount | 0.0100 | 0.0096 | 0.0008 | 0.0001 |

Table 2: Average relative absolute error of algorithms described in this paper and in [DFYTM22; FDY22] on the TPC-H dataset. For each algorithm we executed it 100 times, removed 20 top and 20 bottom values and computed average error for the rest of 60 values.

Most similar to our algorithm is the Race-to-the-Top (R2T) algorithm [DFYTM22]; R2T is a generic framework and the original paper did not specifically consider counting distinct elements, but the approach can easily be applied to $\mathrm{DC}(D; \ell)$. While we use the generalized exponential mechanism [RS15] to select the sensitivity $\ell$, R2T computes multiple lower bounds with different sensitivities $\ell$ and then outputs the maximum of the noisy values. This approach incurs the cost of composition across the multiple evaluations. To manage this cost, R2T only evaluates $\ell = 2, 4, 8, \cdots, 2^{\log \ell_{\max}}$. Compared to our guarantee (8) with an error $O\left(\frac{\ell_*}{\varepsilon} \log\left(\frac{\ell_{\max}}{\beta}\right)\right)$, R2T has a slightly worse theoretical error guarantee of $O\left(\frac{\ell_*}{\varepsilon} \log(\ell_{\max}) \log\left(\frac{\log \ell_{\max}}{\beta}\right)\right)$ [DFYTM22, Theorem 5.1].

The shifted inverse mechanism [FDY22] takes a different approach to the problem. Rather than relying on adding Laplace noise (as we do), it applies the exponential mechanism with an ingenious loss function (see [Ste23] for additional discussion). When applied to counting distinct elements, the shifted inverse mechanism gives an accuracy guarantee comparable to ours (8). The downside of the shifted inverse mechanism is that computing the loss function is, in general, NP-hard. Fang, Dong, and Yi [FDY22] propose polynomial-time variants for several specific tasks, including counting distinct elements. However, the algorithm is still relatively slow.

## 3    Technical Background on Differential Privacy

For detailed background on differential privacy, see the survey by Vadhan [Vad17] or the book by Dwork and Roth [DR14]. We briefly define pure DP and some basic mechanisms and results.

---

**Algorithm 3** Generalized Exponential Mechanism [RS15]

---

1: **procedure** GEM($D \in \mathcal{X}^*$; $q_i : \mathcal{X}^* \to \mathbb{R}$ for $i \in [m]$, $\Delta_i > 0$ for $i \in [m]$, $\varepsilon > 0$, $\beta > 0$)
2:     **Require**: $q_i$ has sensitivity $\sup_{\substack{x,x' \in \mathcal{X}^* \\ \text{neighboring}}} |q(x) - q(x')| \leq \Delta_i$ for all $i \in [m]$.
3:     Let $t = \frac{2}{\varepsilon} \log\left(\frac{m}{\beta}\right)$.
4:     **for** $i \in [m]$ **do**
5:         $s_i \leftarrow \min_{j \in [m]} \frac{(q_i(D) - t\Delta_i) - (q_j(D) - t\Delta_j)}{\Delta_i + \Delta_j}$.
6:     **end for**
7:     Sample $\hat{i} \in [m]$ from the Exponential Mechanism using the normalized scores $s_i$; i.e.,

$$\forall i \in [m] \qquad \mathbb{P}\left[\hat{i} = i\right] = \frac{\exp\left(\frac{1}{2}\varepsilon s_i\right)}{\sum_{k \in [m]} \exp\left(\frac{1}{2}\varepsilon s_k\right)}.$$

8:     **return** $\hat{i} \in [m]$.
9: **end procedure**

---

**Definition 3.1** (Differential Privacy (DP) [DMNS06] )**.** *A randomized algorithm* $M : \mathcal{X}^* \to \mathcal{Y}$ *satisfies $\varepsilon$-DP if, for all inputs $D, D' \in \mathcal{X}^*$ differing only by the addition or removal of an element and for all measurable $S \subset \mathcal{Y}$, we have $\mathbb{P}[M(D) \in S] \leq e^\varepsilon \cdot \mathbb{P}[M(D') \in S]$.*

We refer to pairs of inputs that differ only by the addition or removal of one person's data as *neighboring*. Note that it is common to also consider replacement of one person's data; for simplicity, we

| Data Set | Size | | Words per Person | | | Vocabulary Size |
|---|---|---|---|---|---|---|
| | People | Records | Min | Median | Max | |
| Amazon Fashion | 404 | 8533 | 1 | 14.0 | 139 | 1450 |
| Amazon Industrial and Scientific | 11041 | 1446031 | 0 | 86 | 2059 | 36665 |
| Reddit | 223388 | 7117494 | 0 | 18.0 | 1724 | 102835 |
| IMDB | 50000 | 6688844 | 5 | 110.0 | 925 | 98726 |

Table 3: Data sets details.

do not do this. We remark that there are also variants of DP such as approximate DP [DKMMN06] and concentrated DP [DR16; BS16], which quantitatively relax the definition, but these are not relevant in our application. A key property of DP is that it composes and is invariant under postprocessing.

**Lemma 3.2** (Composition & Postprocessing). *Let $M_1 : \mathcal{X}^* \to \mathcal{Y}$ be $\varepsilon_1$-DP. Let $M_2 : \mathcal{X}^* \times \mathcal{Y} \to \mathcal{Z}$ be such that, for all $y \in \mathcal{Y}$, the restriction $M(\cdot, y) : \mathcal{X}^* \to \mathcal{Z}$ is $\varepsilon_2$-DP. Define $M_{12} : \mathcal{X}^* \to \mathcal{Z}$ by $M_{12}(D) = M_2(D, M_1(D))$. Then $M_{12}$ is $(\varepsilon_1 + \varepsilon_2)$-DP.*

A basic DP tool is the Laplace mechanism [DMNS06]. Note that we could also use the *discrete* Laplace mechanism [GRS09; CKS20].

**Lemma 3.3** (Laplace Mechanism). *Let $q : \mathcal{X}^* \to \mathbb{R}$. We say $q$ has sensitivity $\Delta$ if $|q(D) - q(D')| \leq \Delta$ for all neighboring $D, D' \in \mathcal{X}^*$. Define $M : \mathcal{X}^* \to \mathbb{R}$ by $M(D) = q(D) + \mathrm{Lap}(\Delta/\varepsilon)$, where $\mathrm{Lap}(b)$ denotes laplace noise with mean $0$ and variance $2b^2$ – i.e., $\mathbb{P}_{\xi \leftarrow \mathrm{Lap}(b)}[\xi > t] =$ $\mathbb{P}_{\xi \leftarrow \mathrm{Lap}(b)}[\xi < -t] = \frac{1}{2}\exp\left(\frac{t}{b}\right)$ for all $t > 0$. Then $M$ is $\varepsilon$-DP.*

Another fundamental tool for DP is the exponential mechanism [MT07]. It selects the approximately best option from among a set of options, where each option $i$ has a quality function $q_i$ with sensitivity $\Delta$. The following result generalizes the exponential mechanism by allowing each of the quality functions to have a different sensitivity.

**Theorem 3.4** (Generalized Exponential Mechanism [RS15, Theorem 1.4]). *For each $i \in [m]$, let $q_i : \mathcal{X}^* \to \mathbb{R}$ be a query with sensitivity $\Delta_i$. Let $\varepsilon, \beta > 0$. The generalized exponential mechanism ($\mathrm{GEM}(\cdot; \{q_i\}_{i \in [m]}, \{\Delta_i\}_{i \in [m]}, \varepsilon, \beta)$ in Algorithm 3) is $\varepsilon$-DP and has the following utility guarantee. For all $D \in \mathcal{X}^*$, we have*

$$\mathbb{P}_{\hat{i} \leftarrow \mathrm{GEM}(D; \{q_i\}_{i \in [m]}, \{\Delta_i\}_{i \in [m]}, \varepsilon, \beta)} \left[ q_{\hat{i}}(D) \geq \max_{j \in [m]} q_j(D) - \Delta_j \cdot \frac{4}{\varepsilon} \log\left(\frac{m}{\beta}\right) \right] \geq 1 - \beta.$$

## 4 Experimental Results

We empirically validate the performance of our algorithms using data sets of various sizes from different text domains. We focus on the problem of computing vocabulary size with person-level DP. Section 4.1 describes the data sets and Section 4.2 discusses the algorithms we compare.

### 4.1 Datasets

We used four publicly available datasets to assess the accuracy of our algorithms compared to baselines. Two small datasets were used: Amazon Fashion 5-core [NLM19] (reviews of fashion products on Amazon) and Amazon Industrial and Scientific 5-core [NLM19] (reviews of industrial and scientific products on Amazon). Two large data sets were also used: Reddit [She20] (a data set of posts collected from r/AskReddit) and IMDb [N20; MDPHNP11] (a set of movie reviews scraped from IMDb). See details of the datasets in Table 3.

## 4.2 Comparisons

Computing the number of distinct elements using a differentially private mechanism involves two steps: selecting a contribution bound ($\ell$ in our algorithms) and counting the number of distinct elements in a way that restricts each person to only contribute the given number of elements.

**Selection:** We examine four algorithms for determining the contribution limit:

1. Choosing the true maximum person contribution (due to computational restrictions this was only computed for Amazon Fashion data set).

2. Choosing the 90th percentile of person contributions.

3. Choosing the person contribution that exactly maximizes the utility function $q_\ell(D) = \mathrm{DC}(D; \ell) - \frac{\ell}{\varepsilon} \log(\frac{1}{2\beta})$, where $\varepsilon = 1$, and $\beta = 0.001$.

4. Choosing the person contribution that approximately maximizes the utility function using the generalized exponential mechanism with $\epsilon = 1$.

Note that only the last option is differentially private, but we consider the other comparison points nonetheless.

**Counting:** We also consider three algorithms for estimating the number of distinct elements for a given sensitivity bound $\ell$:

1. For each person, we uniformly sample $\ell$ elements without replacement and count the number of distinct elements in the union of the samples.

2. The linear-time greedy algorithm (Algorithm 2) with $\varepsilon = 1$ and $\beta = 0.001$.

3. The matching-based algorithm (Algorithm 1) with $\varepsilon = 1$ and $\beta = 0.001$.

All of these can be converted into DP algorithms by adding Laplace noise to the result.

In all our datasets "true maximum person contribution" and "90th percentile of person contributions" output bounds that are much larger than necessary to obtain true distinct count; hence, we only consider DP versions of the estimation algorithm for these selection algorithms.

## 4.3 Results

Figure 1 shows the dependency of the result on the contribution bound for each of the algorithms for computing the number of distinct elements with fixed person contribution. It is clear that matching and greedy algorithms vastly outperform the sampling approach that is currently used in practice.

Tables 4 to 7 show the performance of algorithms for selecting optimal person contribution bounds on different data sets. For all bound selection algorithms and all data sets, the sampling approach to estimating the distinct count performs much worse than the greedy and matching-based approaches. The greedy approach performs worse than the matching-based approach, but the difference is about 10% for Amazon Fashion and is almost negligible for other data sets since they are much larger. As for the matching-based algorithm, it performs as follows on all the data sets:

1. The algorithm that uses the bound equal to the maximal person contribution overestimates the actual necessary bound. Therefore, we only consider the DP algorithms for counts estimation. It is easy to see that while the median of the estimation is close to the actual distinct count, the amount of noise is somewhat large.

2. The algorithm that uses the bound equal to the 99th percentile of person contributions also overestimates the necessary bound and behaves similarly to the one we just described (though the spread of the noise is a bit smaller).

3. The algorithms that optimize the utility function are considered: one non-private and one private. The non-private algorithm with non-private estimation gives the answer that is very close to the true number of distinct elements. The private algorithm with non-private estimation gives the answer that is worse, but not too much. Finally, the private algorithm with the private estimation gives answers very similar to the results of the non-private estimation.

## Acknowledgments and Disclosure of Funding

We would like to thank Badih Ghazi, Andreas Terzis, and four anonymous reviewers for their constructive feedback and valuable suggestions. We thank Markus Hasenöhrl for helpful discussions, which helped us identify the problem. In addition, we are grateful to Ke Yi, Wei Dong, and Juanru Fang for bringing their related work [DFYTM22; FDY22] to our attention.

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

| Selection | Counting | Person Contribution Bound | | | Distinct Count | | |
|---|---|---|---|---|---|---|---|
| | | 10th PC | Median | 90th PC | 10th PC | Median | 90th PC |
| Max Contrib | DP Sampling | – | 139 | – | 1196.8 | 1407.5 | 1649.1 |
| Max Contrib | DP Greedy | – | 139 | – | 1174.2 | 1439.2 | 1646.5 |
| Max Contrib | DP Matching | – | 139 | – | 1222.2 | 1460.9 | 1631.0 |
| 90th PC Contrib | DP Sampling | – | 48 | – | 1225.4 | 1296.2 | 1377.9 |
| 90th PC Contrib | DP Greedy | – | 48 | – | 1367.0 | 1432.6 | 1516.3 |
| 90th PC Contrib | DP Matching | – | 48 | – | 1365.3 | 1444.7 | 1524.8 |
| Max Utility | Sampling | – | 41 | – | 1247.0 | 1259.0 | 1270.0 |
| Max Utility | Greedy | – | 20 | – | – | 1376 | – |
| Max Utility | Matching | – | 17 | – | – | 1428 | – |
| DP Max Utility | Sampling | 8.9 | 16.0 | 28.0 | 661.6 | 892.5 | 1124.5 |
| DP Max Utility | Greedy | 8.0 | 11.0 | 17.0 | 1148.0 | 1241.0 | 1348.0 |
| DP Max Utility | Matching | 7.0 | 9.0 | 14.0 | 1252.0 | 1317.0 | 1400.0 |
| DP Max Utility | DP Sampling | 9.0 | 16.0 | 27.1 | 702.4 | 899.1 | 1145.1 |
| DP Max Utility | DP Greedy | 8.0 | 10.0 | 19.0 | 1128.5 | 1224.4 | 1370.8 |
| DP Max Utility | DP Matching | 6.9 | 9.0 | 13.1 | 1220.6 | 1319.1 | 1394.2 |

Table 4: Amazon Fashion: the comparison is for $\ell_{\max} = 100$.

| Selection | Counting | Person Contribution Bound | | | Distinct Count | | |
|---|---|---|---|---|---|---|---|
| | | 10th PC | Median | 90th PC | 10th PC | Median | 90th PC |
| 90th PC Contrib | DP Sampling | – | 297 | – | 32458.1 | 32943.8 | 33452.6 |
| 90th PC Contrib | DP Greedy | – | 297 | – | 36270.3 | 36669.5 | 37019.0 |
| 90th PC Contrib | DP Matching | – | 297 | – | 36236.2 | 36651.7 | 37102.7 |
| Max Utility | Sampling | – | 99 | – | 24967.0 | 25039.0 | 25121.2 |
| Max Utility | Greedy | – | 79 | – | – | 36246 | – |
| Max Utility | Matching | – | 42 | – | – | 36364 | – |
| DP Max Utility | Sampling | 85.9 | 96.0 | 99.0 | 23852.8 | 24739.0 | 25049.8 |
| DP Max Utility | Greedy | 34.0 | 49.0 | 66.1 | 35393.0 | 35839.0 | 36116.9 |
| DP Max Utility | Matching | 22.9 | 30.5 | 43.2 | 36026.8 | 36243.5 | 36371.2 |
| DP Max Utility | DP Sampling | 87.0 | 95.0 | 99.0 | 23997.6 | 24701.1 | 25067.7 |
| DP Max Utility | DP Greedy | 32.9 | 47.5 | 68.0 | 35336.6 | 35776.2 | 36136.6 |
| DP Max Utility | DP Matching | 22.0 | 28.0 | 38.0 | 35970.5 | 36198.9 | 36326.7 |

Table 5: Amazon Industrial and Scientific: the comparison is for $\ell_{\max} = 100$.

## A Proofs

**Lemma A.1** (Sensitivity of $\mathrm{DC}(D; \ell)$). *As in Equation* (2)*, for $D \in (\Omega^*)^n$ and $\ell \in \mathbb{N}$, define*

$$\mathrm{DC}(D; \ell) := \max \left\{ \left| \bigcup_{i \in [n]} v_i \right| : \forall i \in [n] \ v_i \subseteq u_i \land |v_i| \leq \ell \right\}.$$

*Let $D, D' \in (\Omega^*)^*$ be neighboring. That is, $D$ and $D'$ differ only by the addition or removal of one entry. Then $|\mathrm{DC}(D; \ell) - \mathrm{DC}(D'; \ell)| \leq \ell$ for all $\ell \in \mathbb{N}$.*

*Proof.* Without loss of generality $D = (u_1, \cdots, u_n) \in (\Omega^*)^n$ and $D' = (u_1, \cdots, u_{n-1}) \in (\Omega^*)^{n-1}$. I.e., $D'$ is $D$ with person $n$ removed. Let $\ell \in \mathbb{N}$.

| Selection | Counting | Person Contribution Bound | | | Distinct Count | | |
|---|---|---|---|---|---|---|---|
| | | 10th PC | Median | 90th PC | 10th PC | Median | 90th PC |
| 90th PC Contrib | DP Sampling | – | 75 | – | 92480.7 | 92654.8 | 92812.1 |
| 90th PC Contrib | DP Greedy | – | 75 | – | 102544.8 | 102665.7 | 102817.7 |
| 90th PC Contrib | DP Matching | – | 75 | – | 102651.1 | 102784.1 | 102907.8 |
| Max Utility | Sampling | – | 99 | – | 95606.9 | 95692.0 | 95750.3 |
| Max Utility | Greedy | – | 52 | – | – | 102543 | – |
| Max Utility | Matching | – | 32 | – | – | 102685 | – |
| DP Max Utility | Sampling | 89.0 | 96.0 | 99.0 | 94549.9 | 95394.5 | 95656.5 |
| DP Max Utility | Greedy | 26.0 | 33.0 | 50.0 | 102015.0 | 102253.0 | 102527.0 |
| DP Max Utility | Matching | 14.0 | 18.5 | 30.0 | 102357.0 | 102501.5 | 102671.0 |
| DP Max Utility | DP Sampling | 88.8 | 96.0 | 99.0 | 94665.2 | 95375.5 | 95693.5 |
| DP Max Utility | DP Greedy | 27.0 | 34.0 | 53.0 | 102053.2 | 102289.6 | 102531.2 |
| DP Max Utility | DP Matching | 14.9 | 18.5 | 28.0 | 102379.7 | 102512.6 | 102643.9 |

Table 6: Reddit: the comparison is for $\ell_{\max} = 100$.

| Selection | Counting | Person Contribution Bound | | | Distinct Count | | |
|---|---|---|---|---|---|---|---|
| | | 10th PC | Median | 90th PC | 10th PC | Median | 90th PC |
| 90th PC Contrib | DP Sampling | – | 238 | – | 95264.5 | 95593.5 | 95966.1 |
| 90th PC Contrib | DP Greedy | – | 238 | – | 98411.0 | 98734.0 | 99120.0 |
| 90th PC Contrib | DP Matching | – | 238 | – | 98354.2 | 98729.4 | 99164.2 |
| Max Utility | Sampling | – | 29 | – | 49907.8 | 50036.5 | 50195.3 |
| Max Utility | Greedy | – | 29 | – | – | 98459 | – |
| Max Utility | Matching | – | 19 | – | – | 98712 | – |
| DP Max Utility | Sampling | 29.0 | 29.0 | 29.0 | 49899.6 | 50070.5 | 50220.9 |
| DP Max Utility | Greedy | 22.0 | 25.0 | 29.0 | 98244.0 | 98364.0 | 98459.0 |
| DP Max Utility | Matching | 13.0 | 16.0 | 21.0 | 98586.0 | 98674.0 | 98721.0 |
| DP Max Utility | DP Sampling | 29.0 | 29.0 | 29.0 | 49924.2 | 50053.7 | 50211.9 |
| DP Max Utility | DP Greedy | 20.0 | 26.0 | 29.0 | 98126.7 | 98369.6 | 98451.8 |
| DP Max Utility | DP Matching | 12.0 | 16.0 | 21.0 | 98555.6 | 98670.4 | 98726.8 |

Table 7: IMDB: the comparison is for $\ell_{\max} = 30$.

Let $v_1 \subseteq u_1, \cdots, v_n \subseteq u_n$ and $v'_1 \subseteq u_1, \cdots, v'_{n-1} \subseteq u_{n-1}$ satisfy

$$\forall i \in [n] \; |v_i| \leq \ell \quad \text{and} \quad \left| \bigcup_{i \in [n]} v_i \right| = \mathrm{DC}(D; \ell)$$

and

$$\forall i \in [n-1] \; |v'_i| \leq \ell \quad \text{and} \quad \left| \bigcup_{i \in [n-1]} v'_i \right| = \mathrm{DC}(D'; \ell).$$

We can convert the "witness" $v'_1, \cdots, v'_{n-1}$ for $D'$ into a "witness" for $D$ simply by adding the empty set. Define $v'_n = \emptyset$, which satisfies $v'_n \subseteq u_n$ and $|v'_n| \leq \ell$. Then

$$
\mathrm{DC}(D;\ell) = \max \left\{ \left| \bigcup_{i \in [n]} v_i \right| : \forall i \in [n] \ \ v_i \subseteq u_i \wedge |v_i| \leq \ell \right\}
$$

$$
\geq \left| \bigcup_{i \in [n]} v'_i \right|
$$

$$
= \mathrm{DC}(D';\ell).
$$

Similarly, we can convert the "witness" $v_1, \cdots, v_n$ for $D$ into a "witness" for $D'$ simply by discarding $v_n$:

$$
\mathrm{DC}(D';\ell) = \max \left\{ \left| \bigcup_{i \in [n-1]} v'_i \right| : \forall i \in [n-1] \ \ v'_i \subseteq u_i \wedge |v'_i| \leq \ell \right\}
$$

$$
\geq \left| \bigcup_{i \in [n-1]} v_i \right|
$$

$$
\geq \left| \bigcup_{i \in [n]} v_i \right| - |v_n|
$$

$$
\geq \mathrm{DC}(D;\ell) - \ell.
$$

Thus $|\mathrm{DC}(D;\ell) - \mathrm{DC}(D';\ell)| \leq \ell$, as required. $\qquad\square$

*Proof of Theorem 1.1.* **Privacy:** First note that $q_\ell(D) = \mathrm{DC}(D;\ell) - \frac{2\ell}{\varepsilon} \log(1/2\beta)$ has sensitivity $\ell$. Algorithm 1 accesses the dataset via $q_\ell(D)$ in two ways: First it runs the generalized exponential mechanism to select $\hat{\ell}$ and second it computes $\hat{\nu} \leftarrow q_{\hat{\ell}}(D) + \mathrm{Lap}\left(2\hat{\ell}/\varepsilon\right)$. Since the generalized exponential mechanism is $\varepsilon/2$-DP and adding Laplace noise is also $\varepsilon/2$-DP, the overall algorithm is $\varepsilon$-DP by composition.

**Lower bound:** Since $\hat{\nu} \leftarrow q_{\hat{\ell}}(D) + \mathrm{Lap}\left(2\hat{\ell}/\varepsilon\right)$, we have

$$
\mathbb{P}_{\hat{\nu}} \left[ \hat{\nu} \leq q_{\hat{\ell}}(D) + \frac{2\hat{\ell}}{\varepsilon} \log\left(\frac{1}{2\beta}\right) \right] = \mathbb{P}_{\hat{\nu}} \left[ \hat{\nu} \geq q_{\hat{\ell}}(D) - \frac{2\hat{\ell}}{\varepsilon} \log\left(\frac{1}{2\beta}\right) \right] = 1 - \beta. \tag{11}
$$

Since $\mathrm{DC}(D;\hat{\ell}) \leq \mathrm{DC}(D)$ and $q_\ell(D) = \mathrm{DC}(D;\ell) - \frac{2\ell}{\varepsilon} \log(1/2\beta)$, Equation (11) gives

$$
\mathbb{P}_{\hat{\nu}} [\hat{\nu} \leq \mathrm{DC}(D)] \geq \mathbb{P}_{\hat{\nu}} \left[ \hat{\nu} \leq \mathrm{DC}(D;\hat{\ell}) \right] = 1 - \beta.
$$

This is the guarantee of Equation (6); $\hat{\nu}$ is a lower bound on $\mathrm{DC}(D)$ with probability $\geq 1 - \beta$.

**Upper bound:** The accuracy guarantee of the generalized exponential mechanism (Theorem 3.4) is

$$
\mathbb{P}_{\hat{\ell}} \left[ q_{\hat{\ell}}(D) \geq \max_{\ell \in [\ell_{\max}]} q_\ell(D) - \ell \cdot \frac{4}{\varepsilon/2} \log(\ell_{\max}/\beta) \right] \geq 1 - \beta. \tag{12}
$$

Combining Equations (11) and (12) with a union bound yields

$$
\mathbb{P}_{(\hat{\ell},\hat{\nu}) \leftarrow \mathcal{M}(D)} \left[ \hat{\nu} \geq \max_{\ell \in [\ell_{\max}]} \mathrm{DC}(D;\ell) - \frac{2\ell + 2\hat{\ell}}{\varepsilon} \log\left(\frac{1}{2\beta}\right) - \frac{8\ell}{\varepsilon} \log\left(\frac{\ell_{\max}}{\beta}\right) \right] \geq 1 - 2\beta. \tag{13}
$$

To interpret Equation (13) we need a high-probability upper bound on $\hat{\ell}$. Let $A > 0$ be determined later and define

$$
\ell_A^* := \arg\max_{\ell \in [\ell_{\max}]} \mathrm{DC}(D;\ell) - \frac{A\ell}{\varepsilon}, \tag{14}
$$

so that $\mathrm{DC}(D;\ell) \leq \mathrm{DC}(D;\ell_A^*) + (\ell - \ell_A^*)\frac{A}{\varepsilon}$ for all $\ell \in [\ell_{\max}]$.

Assume the event in Equation (12) happens. We have

$$\mathrm{DC}(D;\hat{\ell}) - \frac{2\hat{\ell}}{\varepsilon}\log\left(\frac{1}{2\beta}\right) \leq \mathrm{DC}(D;\ell_A^*) + (\hat{\ell} - \ell_A^*)\frac{A}{\varepsilon} - \frac{2\hat{\ell}}{\varepsilon}\log\left(\frac{1}{2\beta}\right),$$
$$\text{(by Equation (14))}$$

$$
\begin{aligned}
\mathrm{DC}(D;\hat{\ell}) - \frac{2\hat{\ell}}{\varepsilon}\log\left(\frac{1}{2\beta}\right) = q_{\hat{\ell}}(D) &\geq \max_{\ell \in [\ell_{\max}]} q_\ell(D) - \ell \cdot \frac{4}{\varepsilon/2}\log(\ell_{\max}/\beta) \quad \text{(by assumption)}\\
&= \max_{\ell \in [\ell_{\max}]} \mathrm{DC}(D;\ell) - \frac{2\ell}{\varepsilon}\log\left(\frac{1}{2\beta}\right) - \frac{8\ell}{\varepsilon}\log\left(\frac{\ell_{\max}}{\beta}\right)\\
&\geq \mathrm{DC}(D;\ell_A^*) - \frac{2\ell_A^*}{\varepsilon}\log\left(\frac{1}{2\beta}\right) - \frac{8\ell_A^*}{\varepsilon}\log\left(\frac{\ell_{\max}}{\beta}\right).
\end{aligned}
$$

Combining inequalities yields

$$\mathrm{DC}(D;\ell_A^*) - \frac{2\ell_A^*}{\varepsilon}\log\left(\frac{1}{2\beta}\right) - \frac{8\ell_A^*}{\varepsilon}\log\left(\frac{\ell_{\max}}{\beta}\right) \leq \mathrm{DC}(D;\ell_A^*) + (\hat{\ell} - \ell_A^*)\frac{A}{\varepsilon} - \frac{2\hat{\ell}}{\varepsilon}\log\left(\frac{1}{2\beta}\right),$$
$$(15)$$

which simplifies to

$$\hat{\ell} \cdot \left(2\log\left(\frac{1}{2\beta}\right) - A\right) \leq \ell_A^* \cdot \left(2\log\left(\frac{1}{2\beta}\right) + 8\log\left(\frac{\ell_{\max}}{\beta}\right) - A\right). \tag{16}$$

Now we set $A = \log\left(\frac{1}{2\beta}\right)$ to obtain

$$\hat{\ell} \cdot \log\left(\frac{1}{2\beta}\right) \leq \ell_A^* \cdot \left(\log\left(\frac{1}{2\beta}\right) + 8\log\left(\frac{\ell_{\max}}{\beta}\right)\right). \tag{17}$$

Substituting Equation (17) into Equation (13) gives

$$\mathop{\mathbb{P}}_{(\hat{\ell},\hat{\nu})\leftarrow\mathcal{M}(D)}\left[\hat{\nu} \geq \max_{\ell \in [\ell_{\max}]} \mathrm{DC}(D;\ell) - \frac{2\ell + 2\ell_A^*}{\varepsilon}\log\left(\frac{1}{2\beta}\right) - \frac{8\ell + 16\ell_A^*}{\varepsilon}\log\left(\frac{\ell_{\max}}{\beta}\right)\right] \geq 1 - 2\beta. \tag{18}$$

We simplify Equation (18) using $\log\left(\frac{1}{2\beta}\right) \leq \log\left(\frac{\ell_{\max}}{\beta}\right)$ to obtain Equation (7).

**Computational efficiency:** Finally, the runtime of $\mathrm{DPDISTINCTCOUNT}(D)$ is dominated by $\ell_{\max}$ calls to the $\mathrm{SENSITIVEDISTINCTCOUNT}(D)$ subroutine, which computes the maximum size of a bipartite matching on a graph with $|E| = \sum_{i\in[n]} |u_i| \cdot \min\{\ell, |u_i|\} \leq |D| \cdot \ell_{\max}$ edges and $|V| + |U| = \mathrm{DC}(D) + \sum_{i\in[n]} \min\{\ell, |u_i|\} \leq 2|D|$ vertices. The Hopcroft-Karp-Karzanov algorithm runs in time $O(|E| \cdot \sqrt{|V| + |U|}) \leq O(|D|^{1.5} \cdot \ell_{\max})$ time. $\qquad\square$

*Proof of Theorem 1.2.* **Privacy:** For convenience we analyze Algorithm 4, which produces the same result as Algorithm 2. (The difference is that Algorithm 4 is written to be efficient, while Algorithm 2 is written in a redundant manner to make the subroutine we are analyzing clear.) The privacy of the algorithm follows by composing the privacy guarantees of the generalized exponential mechanism and Laplace noise addition. The only missing part is to prove that $\mathrm{SENSITIVEAPPROXDISTINCTCOUNT}(\cdot, \ell)$ has sensitivity $\ell$.

Note that

$$
\begin{aligned}
&\mathrm{SENSITIVEAPPROXDISTINCTCOUNT}((u_1, \cdots, u_n), \ell)\\
&\quad = \mathrm{SENSITIVEAPPROXDISTINCTCOUNT}((\underbrace{u_1, \cdots, u_n, \cdots, u_1, \cdots, u_n}_{\ell \text{ times}}), 1);
\end{aligned}
$$

therefore, it is enough to prove that $\mathrm{SENSITIVEAPPROXDISTINCTCOUNT}(\cdot, 1)$ has sensitivity 1.

Assume $D = (u_1, \cdots, u_n)$ and $D' = (u_1, \cdots, u_{j-1}, u_{j+1}, \cdots, u_n)$. Let $S_1, \cdots, S_n$ and $v_1, \cdots, v_n$ be the states of $S$ and $v$ ($v_i = \bot$ if $u_i \setminus S_{i-1} = \emptyset$), respectively, when

---

**Algorithm 4** Approximate Distinct Count Algorithm

---

1: **procedure** SENSITIVEAPPROXDISTINCTCOUNT($D = (u_1, \cdots, u_n) \in (\Omega^*)^n; \ell \in \mathbb{N}$)
2: $\quad$ $S \leftarrow \emptyset$.
3: $\quad$ **for** $\ell' \in [\ell]$ **do**
4: $\qquad$ **for** $i \in [n]$ with $u_i \setminus S \neq \emptyset$ **do**
5: $\qquad\quad$ Choose lexicographically first $v \in u_i \setminus S$. $\qquad\qquad\qquad$ ▷ Match $(i, \ell')$ to $v$.
6: $\qquad\quad$ Update $S \leftarrow S \cup \{v\}$.
7: $\qquad$ **end for**
8: $\quad$ **end for**
9: $\quad$ **return** $|S|$
10: **end procedure**
11: **procedure** DPAPPROXDISTINCTCOUNT($D = (u_1, \cdots, u_n) \in (\Omega^*)^n; \ell_{\max} \in \mathbb{N}, \varepsilon > 0, \beta \in (0, \frac{1}{2})$)
12: $\quad$ **for** $\ell \in [\ell_{\max}]$ **do**
13: $\qquad$ Define $q_\ell(D) := $ SENSITIVEAPPROXDISTINCTCOUNT$(D; \ell) - \frac{2\ell}{\varepsilon} \cdot \log\left(\frac{1}{2\beta}\right)$.
14: $\quad$ **end for**
15: $\quad$ $\hat{\ell} \leftarrow$ GEM$(D; \{q_\ell\}_{\ell \in [\ell_{\max}]}, \{\ell\}_{\ell \in [\ell_{\max}]}, \varepsilon/2, \beta)$. $\qquad\qquad\qquad$ ▷ Algorithm 3
16: $\quad$ $\hat{\nu} \leftarrow q_{\hat{\ell}}(D) + $ Lap$\left(2\hat{\ell}/\varepsilon\right)$.
17: $\quad$ **return** $(\hat{\ell}, \hat{\nu}) \in [\ell_{\max}] \times \mathbb{R}$.
18: **end procedure**

---

we run APPROXSENSITIVEDISTINCTCOUNT$(D, 1)$. Similarly, let $S'_1, \cdots, S'_{j-1}, S'_{j+1}, \cdots, S'_n$, $v'_1, \cdots, v'_{j-1}, v'_{j+1}, \cdots, v'_n$ be the states of $S$ and $v$ ($v'_i = \perp$ if $u_i \setminus S'_{i-1} = \emptyset$),[6] respectively, when we run APPROXSENSITIVEDISTINCTCOUNT$(D', 1)$. Our goal is to show that $||S_n| - |S'_n|| \leq 1$. Define $S'_j = S'_{j-1}$ and $v'_j = \perp$. Clearly $S'_i = S_i$ and $v'_i = v_i$ for all $i < j$. For $i \geq j$, we claim that $S'_i \subseteq S_i$ and $|S_i| \leq |S'_i| + 1$. This is true for $i = j$, since $S'_j = S'_{j-1} = S_{j-1}$ and $S_j = S_{j-1} \cup \{v_j\}$. We prove the claim for $i > j$ by induction. I.e., assume $S_{i-1} = S'_{i-1} \cup \{v^*_{i-1}\}$ for some $v^*_{i-1}$ (possibly $v^*_{i-1} = \perp$, whence $S_{i-1} = S'_{i-1}$). Now $v_i$ is the lexicographically first element of $u_i \setminus S_{i-1}$ and $v'_i$ is the lexicographically first element of $u_i \setminus S'_{i-1}$ (or $\perp$ if these sets are empty). By the induction assumption, $u_i \setminus S_{i-1} \subseteq u_i \setminus S'_{i-1} \subseteq (u_i \setminus S_{i-1}) \cup \{v^*_{i-1}\}$. Thus either $v'_i = v_i$ or $v'_i = v^*_{i-1}$. If $v'_i = v_i$, then $S'_i = S'_{i-1} \cup \{v'_i\} \subset S_{i-1} \cup \{v_i\} = S_i$ and $S'_i \cup \{v^*_{i-1}\} = S'_{i-1} \cup \{v'_i, v^*_{i-1}\} \supset S_{i-1} \cup \{v_i\} = S_i$. If $v'_i = v^*_{i-1}$, then $S'_i = S'_{i-1} \cup \{v^*_{i-1}\} \subset S_{i-1} \cup \{v^*_{i-1}\} = S_{i-1} \subset S_i$ and $S'_i \cup \{v_i\} = S'_{i-1} \cup \{v_i, v^*_{i-1}\} \supset S_{i-1} \cup \{v_i\} = S_i$, so the claim holds with $v^*_i = v_i$.

The sensitivity bound implies that $q_\ell(D) = |S| - \frac{2\ell}{\varepsilon} \log(1/2\beta)$ has sensitivity $\ell$. Since the generalized exponential mechanism is $\varepsilon/2$-DP and adding Laplace noise is also $\varepsilon/2$-DP, the overall algorithm is $\varepsilon$-DP by composition.

**Lower bound:** Let us denote by $\widehat{\text{DC}}(D; \ell) = $ SENSITIVEAPPROXDISTINCTCOUNT$(\cdot, \ell)$ the value of $|S|$ we obtain on Line 8 in Algorithm 2. Note that $\widehat{\text{DC}}(D; \ell)$ is the size of a maximal matching in $G_\ell$, where $G_\ell$ is the bipartite graph corresponding to the input $D$ with $\ell$ copies of each person (see Algorithm 1 for a formal description of the graph). Since a maximal matching is a 2-approximation to a maximum matching [God], we have

$$\frac{1}{2}\text{DC}(D; \ell) \leq \widehat{\text{DC}}(D; \ell) \leq \text{DC}(D; \ell) \leq \text{DC}(D). \tag{19}$$

Since $\hat{\nu} \leftarrow q_{\hat{\ell}}(D) + $ Lap$\left(2\hat{\ell}/\varepsilon\right)$, we have

$$\mathbb{P}_{\hat{\nu}}\left[\hat{\nu} \leq q_{\hat{\ell}}(D) + \frac{2\hat{\ell}}{\varepsilon} \log\left(\frac{1}{2\beta}\right)\right] = \mathbb{P}_{\hat{\nu}}\left[\hat{\nu} \geq q_{\hat{\ell}}(D) - \frac{2\hat{\ell}}{\varepsilon} \log\left(\frac{1}{2\beta}\right)\right] = 1 - \beta. \tag{20}$$

---

[6]For notational convenience, we define $\{\perp\} = \emptyset$.

Substituting $q_\ell(D) = \widehat{\mathrm{DC}}(D;\ell) - \frac{2\ell}{\varepsilon}\log(1/2\beta)$ into Equations (19) and (20) gives Equation (9)

$$\mathop{\mathbb{P}}_{\hat{\nu}}\left[\hat{\nu} \le \mathrm{DC}(D)\right] \ge \mathop{\mathbb{P}}_{\hat{\nu}}\left[\hat{\nu} \le \widehat{\mathrm{DC}}(D;\hat{\ell})\right] = 1 - \beta.$$

**Upper bound:** The accuracy guarantee of the generalized exponential mechanism (Theorem 3.4) is

$$\mathop{\mathbb{P}}_{\hat{\ell}}\left[q_{\hat{\ell}}(D) \ge \max_{\ell \in [\ell_{\max}]} q_\ell(D) - \ell \cdot \frac{4}{\varepsilon/2}\log(\ell_{\max}/\beta)\right] \ge 1 - \beta. \tag{21}$$

Combining Equations (20) and (21) and the definition of $q_\ell(D)$ with a union bound yields

$$\mathop{\mathbb{P}}_{(\hat{\ell},\hat{\nu}) \leftarrow \mathcal{M}(D)}\left[\hat{\nu} \ge \max_{\ell \in [\ell_{\max}]} \widehat{\mathrm{DC}}(D;\ell) - \frac{2\ell + 2\hat{\ell}}{\varepsilon}\log\left(\frac{1}{2\beta}\right) - \frac{8\ell}{\varepsilon}\log\left(\frac{\ell_{\max}}{\beta}\right)\right] \ge 1 - 2\beta. \tag{22}$$

Now we assume $\max_{i \in [n]} |u_i| \le \ell_* \le \ell_{\max}$ for some $\ell_*$. This implies $\mathrm{DC}(D) = \mathrm{DC}(D;\ell)$. We have

$$\max_{\ell \in [\ell_{\max}]} \widehat{\mathrm{DC}}(D;\ell) - \frac{2\ell + 2\hat{\ell}}{\varepsilon}\log\left(\frac{1}{2\beta}\right) - \frac{8\ell}{\varepsilon}\log\left(\frac{\ell_{\max}}{\beta}\right) \tag{23}$$

$$\ge \widehat{\mathrm{DC}}(D;\ell_*) - \frac{2\ell_* + 2\hat{\ell}}{\varepsilon}\log\left(\frac{1}{2\beta}\right) - \frac{8\ell_*}{\varepsilon}\log\left(\frac{\ell_{\max}}{\beta}\right) \tag{24}$$

$$\ge \frac{1}{2}\mathrm{DC}(D) - \frac{2\ell_* + 2\hat{\ell}}{\varepsilon}\log\left(\frac{1}{2\beta}\right) - \frac{8\ell_*}{\varepsilon}\log\left(\frac{\ell_{\max}}{\beta}\right). \tag{25}$$

As in the proof of Theorem 1.1, if the event in Equation (21) happens, we can show that

$$\hat{\ell} \cdot \log\left(\frac{1}{2\beta}\right) \le \ell_A^* \cdot \left(\log\left(\frac{1}{2\beta}\right) + 8\log\left(\frac{\ell_{\max}}{\beta}\right)\right), \tag{26}$$

where

$$\ell_A^* := \arg\max_{\ell \in [\ell_{\max}]} \widehat{\mathrm{DC}}(D;\ell) - \frac{\ell}{\varepsilon}\log\left(\frac{1}{2\beta}\right) \tag{27}$$

Note that $\ell_A^* \le \ell_*$, since $\widehat{\mathrm{DC}}(D;\ell) \le \widehat{\mathrm{DC}}(D;\ell_*)$ for all $\ell \in [\ell_{\max}]$. (That is simply to say that the size of the maximal matching cannot be increased by making more copies of a vertex once there is one copy for each neighbor.) Combining bounds yields

$$\mathop{\mathbb{P}}_{(\hat{\ell},\hat{\nu}) \leftarrow \mathcal{M}(D)}\left[\hat{\nu} \ge \frac{1}{2}\mathrm{DC}(D) - \frac{4\ell_*}{\varepsilon}\log\left(\frac{1}{2\beta}\right) - \frac{24\ell_*}{\varepsilon}\log\left(\frac{\ell_{\max}}{\beta}\right)\right] \ge 1 - 2\beta. \tag{28}$$

Since $\log\left(\frac{1}{2\beta}\right) \le \log\left(\frac{\ell_{\max}}{\beta}\right)$, this implies Equation (10).

**Computational efficiency:** It only remains to verify that Algorithm 2 can be implemented in $O(|D| + \ell_{\max}\log\ell_{\max})$ time. We can implement $S$ using a hash table to ensure that we can add an element or query membership of an element in constant time. (We can easily maintain a counter for the size of $S$.) We assume $D$ is presented as a linked list of linked lists representing each $u_i$ and furthermore that the linked lists $u_i$ are sorted in lexicographic order. The outer loop proceeds through the linked list for $D = (u_1, \cdots, u_n)$. For each $u_i$, we simply pop elements from the linked list and check if they are in $S$ until either we find $v \in u_i \setminus S$ (and add $v$ to $S$) or $u_i$ becomes empty (in which case we remove it from the linked list for $D$.) Since each iteration decrements $|D|$, the runtime of the main loop is $O(|D|)$. Running the generalized exponential mechanism (Algorithm 3) takes $O(\ell_{\max}\log\ell_{\max})$ time. $\qquad\square$

# B  Implementing the Generalized Exponential Mechanism

We conclude with some remarks about implementing the generalized exponential mechanism of Raskhodnikova and Smith [RS15] given in Algorithm 3. There are two parts to this; first we must compute the normalized scores and then we must run the standard exponential mechanism.

Implementing the standard exponential mechanism is well-studied [Ilv20] and can be performed in linear time (with some caveats about randomness and precision). We remark that, instead of the exponential mechanism, we can use report-noisy-max or permute-and-flip [MS20; DKSWXZ+21; DDKPWWXZ23]. These variants may be easier to implement and may provide better utility too.

The normalized scores are given by

$$\forall i \in [m] \quad s_i = \min_{j \in [m]} \frac{(q_i(D) - t\Delta_i) - (q_j(D) - t\Delta_j)}{\Delta_i + \Delta_j}, \tag{29}$$

where $q_i(D)$ is the value of the query on the dataset, $\Delta_i > 0$ is the sensitivity of $q_i$, and $t$ is a constant. Naïvely it would take $\Theta(m^2)$ time to compute all the normalized scores. However, a more complex algorithm can compute the scores in $O(m \log m)$ time.

Observe that, for each $i \in [m]$, we have

$$s_i = \min_{j \in [m]} \frac{(q_i(D) - t\Delta_i) - (q_j(D) - t\Delta_j)}{\Delta_i + \Delta_j} \iff \min_{j \in [m]} \frac{(q_i(D) - t\Delta_i) - (q_j(D) - t\Delta_j) - s_i(\Delta_i + \Delta_j)}{\Delta_i + \Delta_j} = 0$$

$$\iff \min_{j \in [m]} (q_i(D) - t\Delta_i) - (q_j(D) - t\Delta_j) - s_i(\Delta_i + \Delta_j) = 0$$

$$\iff q_i(D) - (s_i + t)\Delta_i = \underbrace{\max_{j \in [m]} q_j(D) + (s_i - t)\Delta_j}_{f(s_i - t)}.$$

That is, we can compute $s_i$ by solving the equation $q_i(D) - (s_i + t)\Delta_i = f(s_i - t)$.

Since $f(x) := \max_{j \in [m]} q_j(D) + x\Delta_j$ is the maximum of increasing linear functions, we have that $f$ is a convex, increasing, piecewise-linear function, with at most $m$ pieces. We can represent $f$ by a sorted list of the points where the linear pieces connect, along with the linear function on each of the pieces. We can compute this representation of $f$ as a pre-processing step in $O(m \log m)$ time; we sort the lines $y(x) = q_j(D) + x\Delta_j$ by their slope $\Delta_j$ and then compute the intersection points between consecutive lines (we delete lines that never realize the max).

Given the above representation of $f$ and the values $q_i(D), \Delta_i, t$, we can compute $s_i$ in $O(\log m)$ time. We must solve $q_i(D) - (s_i + t)\Delta_i = f(s_i - t)$ for $s_i$. We can perform binary search on the pieces of $f$ to identify $j$ such that $f(s_i - t) = q_j(D) + (s_i - t)\Delta_j$. Once we have this we can directly compute $s_i = \frac{(q_i(D) - t\Delta_i) - (q_j(D) - t\Delta_j)}{\Delta_i + \Delta_j}$. The binary search takes $O(\log m)$ time and we must compute $m$ scores. Thus the overall runtime (including the pre-processing) is $O(m \log m)$.

