# Privately Counting Unique Elements

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

 approximation schemes for counting distinct elements. Desfontaines et al. [2019] proved that a number of existing approximate algorithms allow an attacker to test whether a particular individual is in the collection; therefore, creation of a differentially private scheme requires care. Nonetheless, Smith et al. [2020] proved that Flajolet-Martin Sketch is private by itself and Dickens et al. [2022] proved that several other cardinality estimators can be tweaked to make them private. In case of local and shuffle models the only known results are communication complexity bounds [Chen et al., 2021]. Counting unique elements has been considered in the streaming setting [Dwork et al., 2010, Ghazi et al., 2023].

A closely related problem is that of identifying as many elements as possible (rather than just counting them); this is known as " partition selection," "set union," and "key selection" [Swanberg et al., 2023, Desfontaines et al., 2022, Korolova et al., 2009, Carvalho et al., 2022, Rivera Cardoso and Rogers, 2022, Gopi et al., 2020, Zhang et al., 2023]. Note that, by design, DP prevents us from identifying elements that only appear once in the dataset, or only a few times. Thus we can only output items that appear frequently.

For our problem of counting the number of unique elements under person-level/user-level privacy, the only known algorithm is the algorithm where each user independently samples a subset of their elements to reduce the sensitivity. We use this as a baseline in our experiments and show that our algorithm outperforms it.

# 3   Technical Background on Differential Privacy

For detailed background on differential privacy, see the survey by Vadhan [2017] or the book by Dwork and Roth [2014]. We briefly define pure DP and some basic mechanisms and results.

---

**Algorithm 3** Generalized Exponential Mechanism [Raskhodnikova and Smith, 2015]

---

1: **procedure** GEM($D \in \mathcal{X}^*$; $q_i : \mathcal{X}^* \to \mathbb{R}$ for $i \in [m]$, $\Delta_i > 0$ for $i \in [m]$, $\varepsilon > 0$, $\beta > 0$)
2:     **Require**: $q_i$ has sensitivity $\sup_{\substack{x,x' \in \mathcal{X}^* \\ \text{neighboring}}} |q(x) - q(x')| \le \Delta_i$ for all $i \in [m]$.
3:     Let $t = \frac{2}{\varepsilon} \log \left( \frac{m}{\beta} \right)$.
4:     **for** $i \in [m]$ **do**
5:         $s_i \leftarrow \min_{j \in [m]} \frac{(q_i(D) - t\Delta_i) - (q_j(D) - t\Delta_j)}{\Delta_i + \Delta_j}$.
6:     **end for**
7:     Sample $\hat{i} \in [m]$ from the Exponential Mechanism using the normalized scores $s_i$; i.e.,

$$\forall i \in [m] \qquad \mathbb{P}\left[\hat{i} = i\right] = \frac{\exp\left(\frac{1}{2}\varepsilon s_i\right)}{\sum_{k \in [m]} \exp\left(\frac{1}{2}\varepsilon s_k\right)}.$$

8:     **return** $\hat{i} \in [m]$.
9: **end procedure**

---

**Definition 3.1** (Differential Privacy (DP) [Dwork et al., 2006b] ). *A randomized algorithm* $M$ : $\mathcal{X}^* \to \mathcal{Y}$ *satisfies* $\varepsilon$-*DP if, for all inputs* $D, D' \in \mathcal{X}^*$ *differing only by the addition or removal of an element and for all measurable* $S \subset \mathcal{Y}$, *we have* $\mathbb{P}\left[M(D) \in S\right] \le e^{\varepsilon} \cdot \mathbb{P}\left[M(D') \in S\right]$.

We refer to pairs of inputs that differ only by the addition or removal of one person's data as *neighboring*. Note that it is common to also consider replacement of one person's data; for simplicity, we do not do this. We remark that there are also variants of DP such as approximate DP [Dwork et al., 2006a] and concentrated DP [Dwork and Rothblum, 2016, Bun and Steinke, 2016], which quantitatively relax the definition, but these are not relevant in our application. A key property of DP is that it composes and is invariant under postprocessing.

**Lemma 3.2** (Composition & Postprocessing). *Let* $M_1 : \mathcal{X}^* \to \mathcal{Y}$ *be* $\varepsilon_1$-*DP. Let* $M_2 : \mathcal{X}^* \times \mathcal{Y} \to \mathcal{Z}$ *be such that, for all* $y \in \mathcal{Y}$, *the restriction* $M(\cdot, y) : \mathcal{X}^* \to \mathcal{Z}$ *is* $\varepsilon_2$-*DP. Define* $M_{12} : \mathcal{X}^* \to \mathcal{Z}$ *by* $M_{12}(D) = M_2(D, M_1(D))$. *Then* $M_{12}$ *is* $(\varepsilon_1 + \varepsilon_2)$-*DP.*

A basic DP tool is the Laplace mechanism [Dwork et al., 2006b]. Note that we could also use the *discrete* Laplace mechanism [Ghosh et al., 2009, Canonne et al., 2020].

**Lemma 3.3** (Laplace Mechanism). *Let* $q : \mathcal{X}^* \to \mathbb{R}$. *We say* $q$ *has sensitivity* $\Delta$ *if* $|q(D) - q(D')| \le \Delta$ *for all neighboring* $D, D' \in \mathcal{X}^*$. *Define* $M : \mathcal{X}^* \to \mathbb{R}$ *by* $M(D) = q(D) + \text{Lap}\left(\Delta/\varepsilon\right)$, *where* $\text{Lap}\left(b\right)$ *denotes laplace noise with mean* $0$ *and variance* $2b^2$ – *i.e.,* $\mathbb{P}_{\xi \leftarrow \text{Lap}(b)}\left[\xi > t\right] = \mathbb{P}_{\xi \leftarrow \text{Lap}(b)}\left[\xi < -t\right] = \

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

 1.2.* We start from proving privacy guarantees. Note that Algorithm 2 produces the same result as Algorithm 4. Hence, it is enough to prove that SENSITIVEAPPROXDISTINCTCOUNT$(\cdot, \ell)$ has sensitivity $\ell$. In addition, note that

$$\text{SENSITIVEAPPROXDISTINCTCOUNT}((u_1, \cdots, u_n), \ell) =$$
$$\text{SENSITIVEAPPROXDISTINCTCOUNT}((\underbrace{u_1, \cdots, u_n, \cdots, u_1, \cdots, u_n}_{\ell \text{ times}}), 1);$$

therefore, it is enough to prove that SENSITIVEAPPROXDISTINCTCOUNT$(\cdot, 1)$ has sensitivity 1.

Assume $D' = (u_1, \cdot, u_{j-1}, u_{j+1}, \ldots, u_n)$ and let $S_1, \ldots, S_n, v_1, \ldots, v_n$ be states of $S$ and $v$ ($v_i = \perp$ if $i$ is skipped), respectively, when run APPROXSENSITIVEDISTINCTCOUNT$(D)$ and $S_1', \ldots, S_n', v_1', \ldots, v_n'$ be states of $S$ and $v$ ($v_i = \perp$ if $i$ is skipped), respectively, when run APPROXSENSITIVEDISTINCTCOUNT$(D')$. Let $\{i_1, \ldots, i_k\} = \{i : S_i \neq S_i'\}$. It is clear that $i_1 \geq j$ and $v_{i_1}' = v_j$; similarly $v_{i_2}'$ is either $\perp$ or $v_{i_2}' = v_{i_1}$ etc. As a result $|S_n'| \leq |S_n| \leq |S_n'| + 1$.

The sensitivity bound implies that $q_\ell(D) = |S| - \frac{2\ell}{\varepsilon} \log(1/2\beta)$ has sensitivity $\ell$. Since the generalized exponential mechanism is $\varepsilon/2$-DP and adding Laplace noise is also $\varepsilon/2$-DP, the overall algorithm is $\varepsilon$-DP by composition.

---

**Algorithm 4** Approximate Distinct Count Algorithm

---

 1: **procedure** SENSITIVEAPPROXDISTINCTCOUNT($D = (u_1, \cdots, u_n) \in (\Omega^*)^n; \ell \in \mathbb{N}$)
 2:    $S \leftarrow \emptyset$.
 3:    **for** $\ell' \in [\ell]$ **do**
 4:       **for** $i \in [n]$ with $u_i \setminus S \neq \emptyset$ **do**
 5:          Choose lexicographically first $v \in u_i \setminus S$.                    ▷ Match $(i, \ell)$ to $v$.
 6:          Update $S \leftarrow S \cup \{v\}$.
 7:       **end for**
 8:    **end for**
 9:    **return** $|S|$
10: **end procedure**
11: **procedure** DPAPPROXDISTINCTCOUNT($D = (u_1, \cdots, u_n) \in (\Omega^*)^n; \ell_{\max} \in \mathbb{N}, \varepsilon > 0, \beta \in (0, \frac{1}{2})$)
12:    **for** $\ell \in [\ell_{\max}]$ **do**
13:       Define $q_\ell(D) := $ SENSITIVEAPPROXDISTINCTCOUNT($D; \ell$) $- \frac{2\ell}{\varepsilon} \cdot \log\left(\frac{1}{2\beta}\right)$.
14:    **end for**
15:    $\hat{\ell} \leftarrow$ GEM($D; \{q_\ell\}_{\ell \in [\ell_{\max}]}, \{\ell\}_{\ell \in [\ell_{\max}]}, \varepsilon/2, \beta$).                    ▷ Algorithm 3
16:    $\hat{\nu} \leftarrow q_{\hat{\ell}}(D) + \text{Lap}\left(2\hat{\ell}/\varepsilon\right)$.
17:    **return** $(\hat{\ell}, \hat{\nu}) \in [\ell_{\max}] \times \mathbb{R}$.
18: **end procedure**

---

Let us denote by $\widehat{\text{DC}}(D; \ell)$ the value of $|S|$ we obtain on Line 8 in Algorithm 2. Note that $\widehat{\text{DC}}(D; \ell)$ is size of a maximal matching in $G_\ell$, where $G_\ell$ is the bipartite graph corresponding to the input $D$ with $\ell$ copies of each person (see Algorithm 1 for a formal description of the graph). Since a maximal matching is a 2-approximation to a maximum matching, we have $\widehat{\text{DC}}(D; \ell) \geq \frac{1}{2}\text{DC}(D; \ell)$. Also $\widehat{\text{DC}}(D; \ell) \leq \text{DC}(D; \ell)$.

Since $\hat{\nu} \leftarrow q_{\hat{\ell}}(D) + \text{Lap}\left(2\hat{\ell}/\varepsilon\right)$, we have

$$\mathbb{P}_{\hat{\nu}}\left[\hat{\nu} \leq q_{\hat{\ell}}(D) + \frac{2\hat{\ell}}{\varepsilon} \log\left(\frac{1}{2\beta}\right)\right] = \mathbb{P}_{\hat{\nu}}\left[\hat{\nu} \geq q_{\hat{\ell}}(D) - \frac{2\hat{\ell}}{\varepsilon} \log\left(\frac{1}{2\beta}\right)\right] = 1 - \beta. \qquad (17)$$

Substituting $q_\ell(D) = \widehat{\text{DC}}(D; \ell) - \frac{2\ell}{\varepsilon} \log(1/2\beta)$ into Equation (17) gives

$$\mathbb{P}_{\hat{\nu}}\left[\hat{\nu} \leq \widehat{\text{DC}}(D; \hat{\ell})\right] = \qquad (18)$$

$$\mathbb{P}_{\hat{\nu}}\left[\hat{\nu} \geq \widehat{\text{DC}}(D; \hat{\ell}) - \frac{4\hat{\ell}}{\varepsilon} \log\left(\frac{1}{2\beta}\right)\right] = 1 - \beta. \qquad (19)$$

Combining Equation (18) with $\widehat{\text{DC}}(D; \hat{\ell}) \leq \text{DC}(D)$ yields the guarantee in Equation (6) that $\hat{\nu}$ is a lower bound on $\text{DC}(D)$ with probability $\geq 1 - \beta$.

The accuracy guarantee of the generalized exponential mechanism (Theorem 3.4) is

$$\mathbb{P}_{\hat{\ell}}\left[q_{\hat{\ell}}(D) \geq \max_{\ell \in [\ell_{\max}]} q_\ell(D) - \ell \cdot \frac{4}{\varepsilon/2} \log(\ell_{\max}/\beta)\right] \geq 1 - \beta$$

or, equivalently,

$$\mathbb{P}_{\hat{\ell}}\left[\widehat{\text{DC}}(D; \hat{\ell}) - \frac{2\hat{\ell}}{\varepsilon} \log\left(\frac{1}{2\beta}\right) \geq \max_{\ell \in [\ell_{\max}]} \widehat{\text{DC}}(D; \ell) - \frac{2\ell}{\varepsilon} \log\left(\frac{1}{2\beta}\right) - \frac{8\ell}{\varepsilon} \log\left(\frac{\ell_{\max}}{\beta}\right)\right] \geq 1 - \beta. \qquad (20)$$

Combining Equations (19) and (20) with a union bound yields

$$\mathbb{P}_{(\hat{\ell}, \hat{\nu}) \leftarrow \mathcal{M}(D)}\left[\hat{\nu} \geq \max_{\ell \in [\ell_{\max}]} \widehat{\text{DC}}(D; \ell) - \frac{2\ell + 2\hat{\ell}}{\varepsilon} \log\left(\frac{1}{2\beta}\right) - \frac{8\ell}{\varepsilon} \log\left(\frac{\ell_{\max}}{\beta}\right)\right] \geq 1 - 2\beta. \qquad (21)$$

387 To interpret Equation (21) we need a high-probability upper bound on $\hat{\ell}$. Let $A > 0$ be determined
388 later and define

$$\ell_A^* := \arg\max_{\ell \in [\ell_{\max}]} \widehat{\mathrm{DC}}(D; \ell) - \frac{A\ell}{\varepsilon}, \tag{22}$$

389 so that $\widehat{\mathrm{DC}}(D; \ell) \leq \widehat{\mathrm{DC}}(D; \ell_A^*) + (\ell - \ell_A^*)\frac{A}{\varepsilon}$ for all $\ell \in [\ell_{\max}]$. Assume the event in Equation (20)
390 holds. We have

$$\widehat{\mathrm{DC}}(D; \hat{\ell}) - \frac{2\hat{\ell}}{\varepsilon}\log\left(\frac{1}{2\beta}\right) \leq \widehat{\mathrm{DC}}(D; \ell_A^*) + (\hat{\ell} - \ell_A^*)\frac{A}{\varepsilon} - \frac{2\hat{\ell}}{\varepsilon}\log\left(\frac{1}{2\beta}\right), \quad \text{(by Equation (22))}$$

$$\widehat{\mathrm{DC}}(D; \hat{\ell}) - \frac{2\hat{\ell}}{\varepsilon}\log\left(\frac{1}{2\beta}\right) \geq \max_{\ell \in [\ell_{\max}]} \widehat{\mathrm{DC}}(D; \ell) - \frac{2\ell}{\varepsilon}\log\left(\frac{1}{2\beta}\right) - \frac{8\ell}{\varepsilon}\log\left(\frac{\ell_{\max}}{\beta}\right)$$
$$\text{(by assumption)}$$

$$\geq \widehat{\mathrm{DC}}(D; \ell_A^*) - \frac{2\ell_A^*}{\varepsilon}\log\left(\frac{1}{2\beta}\right) - \frac{8\ell_A^*}{\varepsilon}\log\left(\frac{\ell_{\max}}{\beta}\right).$$

391 Combining inequalities and simplifying yields

$$\hat{\ell} \cdot \left(2\log\left(\frac{1}{2\beta}\right) - A\right) \leq \ell_A^* \cdot \left(2\log\left(\frac{1}{2\beta}\right) + 8\log\left(\frac{\ell_{\max}}{\beta}\right) - A\right). \tag{23}$$

392 Now we set $A = \log\left(\frac{1}{2\beta}\right)$ to obtain

$$\hat{\ell} \cdot \log\left(\frac{1}{2\beta}\right) \leq \ell_A^* \cdot \left(\log\left(\frac{1}{2\beta}\right) + 8\log\left(\frac{\ell_{\max}}{\beta}\right)\right). \tag{24}$$

393 Substituting Equation (15) into Equation (21) gives

$$\mathbb{P}_{(\hat{\ell},\hat{\nu})\leftarrow\mathcal{M}(D)}\left[\hat{\nu} \geq \max_{\ell \in [\ell_{\max}]} \widehat{\mathrm{DC}}(D; \ell) - \frac{2\ell + 2\ell_A^*}{\varepsilon}\log\left(\frac{1}{2\beta}\right) - \frac{8\ell + 16\ell_A^*}{\varepsilon}\log\left(\frac{\ell_{\max}}{\beta}\right)\right] \geq 1 - 2\beta. \tag{25}$$

394 We simplify Equation (25) using $\log\left(\frac{1}{2\beta}\right) \leq \log\left(\frac{\ell_{\max}}{\beta}\right)$ to obtain

$$\mathbb{P}_{(\hat{\ell},\hat{\nu})\leftarrow\widehat{\mathcal{M}}(D)}\left[\hat{\nu} \geq \max_{\ell \in [\ell_{\max}]} \widehat{\mathrm{DC}}(D; \ell) - \frac{10\ell + 18\ell_A^*}{\varepsilon}\log\left(\frac{\ell_{\max}}{\beta}\right)\right] \geq 1 - 2\beta. \tag{26}$$

395 Note that $\widehat{\mathrm{DC}}(D; \ell) \geq \frac{1}{2}\mathrm{DC}(D; \ell)$; hence,

$$\mathbb{P}_{(\hat{\ell},\hat{\nu})\leftarrow\widehat{\mathcal{M}}(D)}\left[\hat{\nu} \geq \max_{\ell \in [\ell_{\max}]} \frac{1}{2}\mathrm{DC}(D; \ell) - \frac{10\ell + 18\ell_A^*}{\varepsilon}\log\left(\frac{\ell_{\max}}{\beta}\right)\right] \geq 1 - 2\beta.$$

396 Finally, note that $\ell_A^* \leq \ell_*$; therefore, we proved Equation (7).

397 It only remains to verify that Algorithm 2 can be implemented in $O(|D|)$ time. We can implement
398 $S$ using a hash table to ensure that we can add an element or query membership of an element in
399 constant time. (We can easily maintain a counter for the size of $S$.) We assume $D$ is presented as
400 a linked list of linked lists representing each $u_i$ and furthermore that the linked lists $u_i$ are sorted
401 in lexicographic order. The outer loop proceeds through the linked list for $D = (u_1, \cdots, u_n)$. For
402 each $u_i$, we simply pop elements from the linked list and check if they are in $S$ until either we find
403 $v \in u_i \setminus S$ (and add $v$ to $S$) or $u_i$ becomes empty (in which case we remove it from the linked list
404 for $D$.) Since each iteration decrements $|D|$, the runtime is $O(|D|)$. $\qquad\square$