# OpenReview forum: "Counting Distinct Elements Under Person-Level Differential Privacy"
_NeurIPS.cc/2023/Conference — NeurIPS 2023 poster_

### Official Review · Reviewer_BQdm · 2023-07-05

**Soundness:** 3 good
**Presentation:** 3 good
**Contribution:** 3 good
**Rating:** 7
**Confidence:** 4

**Summary:**

Suppose a data set consists of (user, item) pairs. The paper provides an estimate of the count of the number of distinct items that satisfies user level differential privacy. This differs from existing work in two ways. Existing work 1) considers item streams with item, not user, level differential privacy and 2) is highly concerned about computation costs and uses data sketches which have bounded size.

It solves the problem by generating lower bounds with bounded sensitivity by converting the distinct count to a max flow problem.

**Strengths:**

The paper introduces a novel user-level privacy formulation of the distinct count problem. The main ideas are presented well, and it is technically sound.

**Weaknesses:**

1. The title could better describe the key feature of the problem, providing user level privacy. Since most distinct counting papers are about approx distinct counting and streaming/distributed algorithms, I think readers have an expectation that this paper would address this given the title and it's a bit of a let down when it doesn't.

2. The related work section is somewhat inaccurate in its descriptions. E.g.
* Desfointaine et al is a universal result about any distinct counting method which is mergeable. Changing the algorithm design can't get around this. But the results only imply that you can't merge many times and preserve both privacy and accuracy, not that you can't build a private sketch.
* Smith et al doesn’t actually analyze the Flajolet-Martin (a.k.a. PCSA) sketch but their own tweak of the LogLog sketch (by Durand and Flajolet). (Though this inaccuracy is totally understandable given their title.)
* Dickens et al is also a universal result for any order invariant sketch.
* Both Smith et al and Dickens et al are pan private in the streaming setting as well
* Note neither claim to provide privacy under merges.

3. There is also some related work on private multiparty computation for distinct counts. e.g. "Privacy-Preserving Secure Cardinality and Frequency Estimation" Kreuter et al 2020

4. The empirical results for the greedy algorithm are nice, but the bound is quite bad for this application since it only gives you an estimate of the correct order of magnitude. It's also not clear what to do in large scale distributed data settings. These are fairly minor weaknesses given the paper's scope though.

**Questions:**

1. What is the distinction between maximal and maximum bipartite matching? A quick search seems to regard these descriptions as the same problem. Does the greedy algorithm generate the maximal matching that approximates the maximum matching?

2. I'd be happy to increase the rating if the weaknesses 1 & especially 2 are addressed.

**Limitations:**

yes

---

> ### Author Rebuttal · Authors · 2023-08-09
>
> We would like to thank the reviewer for their thoughtful comments.
>
> **W1.** We agree that the title could be misleading, we will rename the next revision to “Counting Unique Elements Under Person-level Differential Privacy”.
>
> **W3 & W3.** Thank you very much for pointing out the issues in our related work section, we will make all the suggested changes in the next revision of the paper.
>
> **W4.** Unfortunately, the linear time algorithm’s worst-case guarantees are not great, but it is well-known that the greedy algorithm for approximating maximum matching performs well in practice.  As for the distributed data case, unfortunately we don’t have a solution now: it is known that computing maximum matching in a distributed case is tricky, and it is even more tricky for us since we must bound the sensitivity.
>
> **Q1.** “Maximal matching” and “maximum matching” is the standard terminology (see https://en.wikipedia.org/wiki/Matching_(graph_theory)#Definitions ) though it is understandably confusing.
>
> - A maximal matching is a matching that is maximal by inclusion; i.e., M is a maximal matching iff for any matching M’ such that M subset of M’, M’=M.
> - A maximum matching is a matching that is maximal by size/cardinality; i.e., M is a maximum matching iff for any matching M’ such that |M| <= |M’|, |M’|=|M|.

---

> > ### Comment · Reviewer_BQdm · 2023-08-18
> >
> > Thanks for your response. I've updated by score.
> >
> > Also two additional references are likely of interest. These cover approximate distinct counting where the sketches are mergeable and the hash is public.
> >
> > Efficient Differentially Private F_0 Linear Sketching, Pagh and Stausholm 2020
> > Sketch-Flip-Merge: Mergeable Sketches for Private Distinct Counting, Hehir et al 2023

---

> > > ### Author Response · Authors · 2023-08-18
> > >
> > > Thanks for the score and references; we will definitely add them to the related work section.
> > > Both papers study private distinct counting in a different setting, but they demonstrate that there is plenty of interest in the subject of our work.

---

### Official Review · Reviewer_FDcy · 2023-07-05

**Soundness:** 4 excellent
**Presentation:** 3 good
**Contribution:** 3 good
**Rating:** 6
**Confidence:** 4

**Summary:**

The authors study the fundamental problem of counting the number of distinct elements in a dataset
in a user-level DP setting, where a user can contribute an unbounded number of items. The main
contribution of the paper is an approach to obtain lower bounds through a bounded sensitivity
count and a bias-variance perspective. The authors give interesting algorithms to compute
it quite efficiently, and show a linear time version which trades off efficiency with accuracy.
The algorithms are evaluated empirically and show good performance.

**Strengths:**


The problem is very natural and well motivated. The approach considered in the paper, in terms of
bounded sensitivity, seems quite natural, and is connected to a big literature on Lipschitz extension.
The authors give rigorous bounds on the bounds achieved by their algorithms. The empirical results
are also interesting, and show reasonable performance. The greedy algorithm is also shown to perform
quite well. The presentation is quite good.

**Weaknesses:**

The basic ideas in the paper, and the techniques used in the algorithms are fairly simple. The experiments section could be strengthened

**Questions:**

In line 51, what is the motivation for considering
M_{\ell,\epsilon}(D) - \frac{\ell}{\epsilon}\log(1/2\beta)?

Is it assumed that there is a fixed \ell_{max} for all the data (line 58)? It appears as an input in
procedure DPDISTINCTCOUNT, and in the bounds in Theorem 1.1.  This should be clarified
earlier in that case. The algorithm needs to know \ell_{max} in that case. Is there a way to specify it?

Line 13 of algorithm 1 and line 10 of Algorithm 2: it might be helpful to make GEM consistent with
the name "Generalized Exponential Mechanism" used in Algorithm 3

The baseline sampling algorithm mentioned in lines 131-134 is not really related work, but just a
baseline, and could be in the experiments section.

line 176: "three" should be "four"

line 191: is the sampling done with replacement?

What is the \epsilon value in Fig 1? Does the "Person Contribution Bound" refer to \ell_{max}?
The plots are quite interesting. Is there an explanation for why the gap between the algorithms
and the true counts decreases as in the figure?

Tables 2 and 3 are not very easy to interpret. It might be easier to compare the counts as
fractions or visually.

How do the results vary with \epsilon?

What is the motivation for considering \epsilon=1 in the person contribution for the baseline?

**Limitations:**

To some extent. There are no negative social impacts

---

> ### Author Rebuttal · Authors · 2023-08-09
>
> We would like to thank you for reviewing the paper.
>
> **Q1. Why $M_{\ell,\varepsilon}(D) - \frac{\ell}{\varepsilon}\log(1/2\beta)$?**
> Under user-level DP, it is impossible to compute the exact number of distinct elements since a single user could contribute an arbitrary number of unique elements and differential privacy prevents us from revealing the contribution of a single user.
> Thus our goal is to produce a *lower bound* on the number of distinct elements that is as close as possible to the true count. This is the only meaningful guarantee that we can provide under user-level differential privacy.
> Hence, to ensure that our output is a lower bound, we have to lower the estimate by $\frac{\ell}{\varepsilon}\log(1/2\beta)$ in order to compensate for the possibility that the noise increases the estimate.
>
>
> **Q2. $\ell_\mathrm{max}$** The parameter $\ell_\max$ is a hyper parameter that is provided as input to the algorithm. This is necessary, as there is no a priori way for the algorithm to know how many elements users are contributing.
>
> **Q3. GEM.** Thanks for highlighting this issue, we will change the caption of the algorithm to say “Generalized Exponential Mechanism (GEM)”.
>
> **Q5. Typo.** Thanks for catching this, we will change this in the next revision of the paper.
>
> **Q6. Sampling.** The sampling is without replacement, since it would only reduce the quality of the algorithm to do sampling with replacement.
>
> **Q7. Fig 1.** Figure 1 shows the results without noise, i.e. $\varepsilon=\infty$. The person contribution bound is $\ell$. So the “matching algorithm” is simply plotting $DC(D,\ell)$ as $\ell$ varies, while the other algorithms show how well the baseline and linear-time algorithm can approximate this.
>
> **Q8. Tables 2 & 3.** Thanks for the suggestion! We will explore possibility of splitting the tables or using some sort of bar plots to visualize them.
>
> **Q9. $\varepsilon$ dependence.** See the attached document for the plot showing dependency on epsilon. (It will be a part of the next revision).
>
> **Q10. Why $\varepsilon = 1$.** This choice is arbitrary; we consider 1 to be a reasonable value to evaluate, but we will add further values.

---

> > ### Comment · Reviewer_FDcy · 2023-08-19
> >
> > Thanks to the authors for the detailed responses. They address all my concerns.

---

### Official Review · Reviewer_REV6 · 2023-07-07

**Soundness:** 3 good
**Presentation:** 3 good
**Contribution:** 2 fair
**Rating:** 5
**Confidence:** 3

**Summary:**

Edit: Overall decision recommendation changed from borderline reject to borderline accept under the expectation that limitations and future work directions as discussed in the reviewer discussion are incorporated into the manuscript.

The manuscript studies the problem of approximating the number of unique items in a multiset under (user-level) differential privacy (DP). Unlike in prior work users can contribute arbitrarily many items to the multiset.

The authors propose to select for each user a subset of contributed items that maximises the total number of unique items. The maximisation problem is solved exactly in polynomial-time via bipartite maximum matching or alternatively a 2-approximation is found in linear-time via a greedy algorithm.

**Strengths:**

S1) Non-trivial results: Sensitivity analysis of approximate DC(D;l) and application of generalized exponential mechanism is interesting.

S2) Could be useful: Probability that noisy count is a lower bound can be calibrated and lower bound can be useful to check if certain thresholds are exceeded.

S3) Improvement over naive baselines: Computing/approximating DC(D;l) seems clearly preferable to selecting l random items for each user.

**Weaknesses:**

W1) Heavily biased estimator.

A DP count would seem to prohibit deterministic lower/upper bounds. While a lower bound w.h.p. could still be useful, most settings would seem to call for an unbiased estimator (see also Q1a & Q1b).

W2) Quality of solution unclear.

Apart from the poorly presented tables that do not clearly indicate the correct count, there is also no analysis of how much more utility could potentially be possible for this problem under a given DP regime. Thus, it is difficult to appreciate the solution in terms of how practically useful it is or how much utility is potentially lost due to the method (see also Q2a & Q2b).

W3) Reproducibility

While the datasets seem to be specified, the submission does not seem to provide code for experiments (see also Q3).

W4) Minor: Some parts of the paper are hard to understand. Particularly, Section 3 could focus more on explanations related to the Generalized Exponential Mechanism (GEM) rather than well-known basics.

It would be better to move the basics of DP and basic Exponential/Laplace mechanism to the appendix and instead provide more explanations/intuitions on how the GEM is used in this work, why it is needed and some intuitions how it leads to better results.

W5) Minor: Sensitivity analysis of DC(D;l) is missing.

It could be stated more explicitly why if a user is added that contributes X items DC(D;l) can increase by at most X, i.e., the added user cannot improve the solution for the previous users (e.g., due to optimal solution for previous users). That would also make it clearer why the sensitivity analysis for the approximation of DC(D;l) is non-trivial where an added user could potentially improve the solution for the previous users and removing a user could lead to less optimal solutions for the rest of users.

**Questions:**

Q1a) Presuming number of distinct items per user cannot be arbitrarily large, is it possible to derive an unbiased estimator using the proposed method or is there some strong indication that any unbiased estimator would be inherently impossible (presuming upper limit on items per user) or could not provide good enough accuracy in this setting?

Q1b) Is there some indication that the released noisy counts correlate with the correct counts sufficiently to rank multiple noisy counts somewhat accurately?

Q2a) What is correct count for each table and what is the achieved count with the proposed method?

Q2b) Is there any strong indication that the proposed solution could not be easily improved upon?

Q3) Do the authors plan to release the code that was used for the experiments?

**Limitations:**

Some limitations do not seem to be mentioned (see Weaknesses W1) and there is generally no discussion of limitations in the abstract or an explicit paragraph in the main paper or appendix.

---

> ### Author Rebuttal · Authors · 2023-08-09
>
> We would like to thank the reviewer for their thoughtful comments. We address the questions and weaknesses listed.
>
> **Q1a. -- bias of estimator** Under the assumption that a bound $\ell$ on the number of distinct elements per person is known, it is possible to estimate $DC(D)$ in an unbiased manner by releasing $DC(D, \ell) + Lap(\ell / \varepsilon)$. However, this assumption is very restrictive and, in practice, $\ell$ may be so large that it gives us an estimator with unacceptably large variance. The issue is that the distribution on sizes of $u_i$’s is usually heavy-tailed. Thus we opt for a biased estimator, which provides a clear guarantee – that it is a lower bound with high probability.
>
> **Q1b. – quality of solution** Unfortunately we are not sure that we understand the question. In the worst case the estimate and the true count could be arbitrary different, but in realistic examples they are close to each other.
>
> **Q2a.  – true counts.** We apologize that this was not clear; we will update the results to clarify. Table 1 contains the true counts for each dataset and Tables 2-5 contain the results of the proposed methods; however, the attached table contains the summary with the results returned by the proposed method (for $\varepsilon = 1$, $\beta = 0.05$, and $\ell_\mathrm{max} = 100$).
>
> **Q2b. – possible improvement.** Our intuition is that our algorithm is optimal, but it is not clear how to even state such a claim formally. Roughly, we believe that $DC(D,\ell)$ is the “best” possible estimator for $DC(D)$ that has sensitivity bounded by $\ell$.
>
> **Q3. – reproducibility.** Yes, we plan to release the code with the next revision of the paper. Unfortunately, we were not able to obtain the necessary approval in time for the NeurIPS deadline.
>
> As for weaknesses mentioned, we are going to add more details in the discussion of the Generalized Exponential Mechanism, since this is not well known; however, it is not the novel contribution of our paper so we don’t want to over emphasize it. We will also add a more detailed sensitivity analysis of $DC(D, \ell)$.

---

> > ### Comment · Reviewer_REV6 · 2023-08-17
> >
> > Thank you for the concise and clear answers to the questions! The premise of Q1b is that a biased estimator could still be useful for ranking-type of tasks. More details:
> >
> > - Suppose we have a few datasets $A,B,C,...Z$ with $DC(A) < DC(B) < DC(C) < ... < DC(Z)$ that do not share any users such that we could employ parallel composition.
> > - Suppose that the goal is to establish a ranking of those datasets based on the distinct count ($DC(A) < DC(B) < DC(C) < ... < DC(Z)$) , but we are bound to the DP constraint
> > - What can be said about the expected quality of the ranking based on the biased estimator vs. ranking based on true counts? (e.g. probability of identifying the highest/lowest count if it is larger/smaller than the other counts by a certain amount)

---

> > > ### Author Response · Authors · 2023-08-18
> > >
> > > Thanks for the great question. The short answer is that it depends on the specific datasets. If a single person contributes a huge number of unique items, then this could completely change the ordering, but differential privacy prevents us from being able to reveal this change.
> > >
> > > However, if the datasets $A=(u\_1,\cdots,u\_n)$ and $B=(u’\_1,\cdots,u’\_{n’})$ are directly comparable, then we can argue that the ordering is preserved.
> > > Specifically, suppose that each person/item in $A$ can be mapped to a person/item in $B$ such that if a person has an item in $A$, then the corresponding person also has the corresponding item in $B$. (I.e. under the mapping equivalence $u\_i \subseteq u’\_i$ for each $i \le n \le n’$.)
> > >
> > > This assumption implies $DC(A,\ell) \le DC(B,\ell)$ for all $\ell$.
> > > Our algorithm simply subtracts a constant from and adds noise to this quantity. Thus, the order is preserved, at least in terms of the expectation. The noise could, of course, reverse the order, but this is unavoidable and should only happen when the counts are fairly close.
> > >
> > > Of course, this is a strong assumption. But we expect that if the datasets have similar properties overall, then the order would be preserved.

---

### Official Review · Reviewer_fhMe · 2023-07-08

**Soundness:** 3 good
**Presentation:** 3 good
**Contribution:** 3 good
**Rating:** 5
**Confidence:** 4

**Summary:**


This paper proposes an algorithm to count the maximum number of unique items one can get from considering from every individual in a dataset at most l items, while preserving differential privacy.


**Strengths:**



The paper offers an (as far as I know) original contribution, which is simple and nice.  The explanation is clear.  The result may be significant to the extent one can cast applications needing to count the number of unique items into the relaxation considered in the current paper.
The paper performs an interesting empirical evaluation, even if it doesn't fully succeed to bring up a convincing application.


**Weaknesses:**


The paper also proposes a linear time algorithm, which is very nice, but which makes a second approximation, making it even more unclear how to determine whether it is suitable for a specific application.



**Questions:**

--

**Limitations:**


The paper doesn't have a conclusions section, and doesn't mention limitations nor societal impact.

---

> ### Author Rebuttal · Authors · 2023-08-09
>
> We would like to thank you for reviewing the paper, we respond below.
>
> **Applications:** We believe that applications of our algorithms are abundant. COUNT(DISTINCT …) is a very common SQL aggregation that is missing in all existing differentially private SQL solutions. [A concrete example](https://socialscience.one/blog/unprecedented-facebook-urls-dataset-now-available-research-through-social-science-one) query would be counting the number URLs that a set of users have shared on a platform.
>
> **Suitability of our algorithms:** The motivation for our linear time algorithm is that our main algorithm may not scale to massive datasets. While the linear-time algorithm doesn’t provide strong provable accuracy guarantees, the experiments show that it performs comparable to the optimal algorithm and outperforms the baseline by a large margin.

---

> > ### Comment · Reviewer_fhMe · 2023-08-18
> >
> > Thanks for the rebuttal.
> >
> > I agree that there are many applications for set intersection, but this doesn't mean that in these applications any type of approximation is acceptable.  The comment in my review is that due to *two* approximations it becomes even harder to understand for a specific application whether this algorithm is suitable and if so what approximation parameters would be reasonable.  Moreover, it seems that if we are interested in a linear-time algorithm without caring too much about approximation quality, other algorithms that the one proposed may be simpler.  So my concern here is mainly the lack of analysis of the approximations.

---

> > > ### Author Response · Authors · 2023-08-19
> > >
> > > Thanks for elaborating. We agree that we need to be careful about the approximation.
> > >
> > > What makes it difficult is that, in the setting of person-level differential privacy (a.k.a. user-level DP), a single person can contribute an unbounded number of items. Thus a "normal" approximation guarantee is impossible, because the sensitivity of the true distinct count is infinite.
> > >
> > > We thought a lot about what kind of guarantee we can offer in this setting. We decided that the simplest meaningful guarantee we can give is that, with high probability, the output of our algorithm is $\le$ the true distinct count. So, whatever our algorithm outputs, you can be confident that the true number of distinct items is at least that much.
> > >
> > > Of course, on its own, such a one-sided guarantee is not satisfying.
> > > Intuitively, our algorithm computes the largest possible DP lower bound.
> > > To give a precise guarantee, if we assume that the number of items per person is bounded by $\ell_*$, then the output of our algorithm $\hat\nu$ satisfies $$ DC(D) \ge \hat\nu \ge DC(D) - \widetilde{O}\left(\frac{\ell_*}{\varepsilon}\right) $$ with high probability. (See Equation 5 in the submission for a precise guarantee.)
> > >
> > > ---
> > >
> > > An alternative approach is to provide item-level DP. That is, the differential privacy guarantee assumes that each person contributes only one item (or, more generally, a bounded number of items).
> > > If a person contributes many items, then the assumption is violated and their privacy may degrade arbitrarily.
> > > This was the approach taken for the [Facebook URLs Dataset](https://solomonmg.github.io/pdf/Facebook_DP_URLs_Dataset.pdf).
> > >
> > > Essentially the distinction between our approach and the item-level DP approach is how they handle a person contributing an excessive number of items.
> > > Our approach still provides the desired privacy guarantee, but the accuracy will degrade.
> > > The item-level DP approach provides good accuracy, but the privacy guarantee will fail.
> > > Both approaches can be justified, but we believe that our approach is a valuable contribution to the literature.
> > >
> > > ---
> > >
> > > Thanks again for asking about this important design choice and apologies for the long reply.
> > > We will add further discussion about why we give this style of guarantee to the paper.

---

### Author Rebuttal · Authors · 2023-08-09

Plots that we requested by reviewers are in the attached file.

---

### Decision · Program_Chairs · 2023-09-21

**Decision:**

Accept (poster)

**Comment:**

Summary:
=======
This paper introduces a user-level differentially private method for counting unique elements of a dataset that is shared by users. A user may contribute many elements to this count. The difficulty is that that there is no bound on individual contribution and some users may have very skewed contributions in the tail.

This paper presents two algorithms: a polynomial time exact solution and linear approximation of the $\epsilon$-DP counting algorithm. Theoretical definition and analysis of the algorithm is provided first, alongside evaluation on four datasets. And contributes the user contribution bounds (including 10,50,90%ile) for the datasets.

Reviews:
=========
The paper has borderline reviews. However the high confidence (4) reviewers have given the higher scores (5,6,7) leaning towards accept and the lower confidence (3) reviewers leaning towards the lower scores (4). The authors addressed weaknesses and concerns in review BQdm, resulting in an increase in score to 7.
Reviewers are generally in concordance with understanding the contributions of the paper. However, each reviewer identifies several weaknesses of the work.

* CW1: Application areas
* CW2: Biased estimator
* CW3: Solution quality
* CW4: Simplicity of algorithm
* CW5: Inaccuracy in related work section
* CW6: Novelty of work w.r.t. Kreuter et al., 2020

Some of these are mitigated through discussion:

* CW1: Federated settings, and multi-party counting are application areas where this work can be applied

* CW2: The authors respond with a suitable justification for why biased estimator is not a major issue, providing a formulation of unbiased estimator with discussion justifying why the variance of this estimator would be unacceptable.

* CW3: Empirical results show the generation solutions were acceptable. I assume that modification of $\epsilon$ will allow some tuning of the privacy-accuracy tradeoff. However, this was not studied empirically in the paper.

* CW4: I do not believe this to be a reasonable reason to reject a paper. The underlying task is counting unique elements. The paper uses a DP framework where the contributions of a single party are bounded. This is a reasonable contribution.

* CW5: I believe these issues to be minor and would encourage the authors to follow the corrections from  BQdm.

* CW6: The Wright/Skvortsov/Kreuter/Wang/Mirisola paper does introduce the privacy preserving problem with a DP framework introduced by reviewer BQdm. I would suggest this could be included in the literature survey. Reading the paper, it appears as if this is an introduction of many methods including database-level DP rather than user-level DP and has much less rigorous analysis than the submitted manuscript.




Reason for decision:
========
Reviewing the merits of the paper and in conjunction with the reviews, there are some weaknesses identified with adequate mitigations.  I think it is probably worthy of acceptance, but probably a weaker accept.  Reviewers provided accept ratings of 5,4,6,7 with confidence 4,3,4,4